# Optical Transformers

**Maxwell G. Anderson**                                          *mga58@cornell.edu*
*Department of Applied and Engineering Physics, Cornell University*

**Shi-Yuan Ma**
*Department of Applied and Engineering Physics, Cornell University*

**Tianyu Wang**
*Department of Applied and Engineering Physics, Cornell University*

**Logan G. Wright**
*Department of Applied and Engineering Physics, Cornell University*
*NTT Physics & Informatics Laboratories, NTT Research*

**Peter L. McMahon**                                            *pmcmahon@cornell.edu*
*Department of Applied and Engineering Physics, Cornell University*
*Kavli Institute at Cornell for Nanoscale Science, Cornell University*

**Reviewed on OpenReview:** *https://openreview.net/forum?id=Xxw0edFFQC*

## Abstract

The rapidly increasing size of deep-learning models has renewed interest in alternatives to digital-electronic computers as a means to dramatically reduce the energy cost of running state-of-the-art neural networks. Optical matrix-vector multipliers are best suited to performing computations with very large operands, which suggests that large Transformer models could be a good target for them. In this paper, we investigate—through a combination of simulations and experiments on prototype optical hardware—the feasibility and potential energy benefits of running Transformer models on future optical accelerators that perform matrix-vector multiplication.

We use simulations, with noise models validated by small-scale optical experiments, to show that optical accelerators for matrix-vector multiplication should be able to accurately run a typical Transformer architecture model for language processing. We demonstrate that optical accelerators can achieve the same (or better) perplexity as digital-electronic processors at 8-bit precision, provided that the optical hardware uses sufficiently many photons per inference, which translates directly to a requirement on optical energy per inference. We studied numerically how the requirement on optical energy per inference changes as a function of the Transformer width $d$ and found that the optical energy per multiply–accumulate (MAC) scales approximately as $\frac{1}{d}$, giving an asymptotic advantage over digital systems.

We also analyze the total system energy costs for optical accelerators running Transformers, including both optical and electronic costs, as a function of model size. We predict that well-engineered, large-scale optical hardware should be able to achieve a $100\times$ energy-efficiency advantage over current digital-electronic processors in running some of the largest current Transformer models, and if both the models and the optical hardware are scaled to the quadrillion-parameter regime, optical accelerators could have a $> 8{,}000\times$ energy-efficiency advantage. Under plausible assumptions about future improvements to electronics and Transformer quantization techniques ($5\times$ cheaper memory access, double the digital–analog conversion efficiency, and 4-bit precision), we estimate that the energy advantage for optical processors versus electronic processors operating at 300 fJ/MAC could grow to $> 100{,}000\times$.

# 1   Introduction

Deep learning models' exponentially increasing scale is both a key driver in advancing the state-of-the-art and a cause of growing concern about their energy usage, speed, and practicality. This has led to the development of hardware accelerators and model training/compression/design techniques for efficient and fast inference on them. Because they still perform all the underlying operations using the same physical mechanisms, most digital-electronic accelerators (Reuther et al., 2020; Graphcore, 2021; Cerebras Systems, 2021; Andersch et al., 2022; Habana Labs, 2022) can improve performance by constant factors. While these may be large factors, they do not change the way costs scale with model compute requirements.

Analog accelerators can be different from digital ones in that the energy cost of performing computations may fundamentally scale differently than digital systems. For example, in optics or analog-electronic crossbar arrays, a common heuristic is that the energy of a matrix-vector product scales *linearly* with vector size, rather than the $\sim d^2$ of digital systems (assuming all dimensions are $\sim d$). This is a key intuition for why alternative analog computing platforms using optics have been proposed as a new paradigm for better scalability (Sebastian et al., 2020; Caulfield & Dolev, 2010; Wetzstein et al., 2020; Nahmias et al., 2020; Stark et al., 2020; Huang et al., 2021; Shastri et al., 2021). Ideally, the scaling is asymptotically better than digital systems in energy per MAC (Hamerly et al., 2019; Wang et al., 2022; Sludds et al., 2022; Nahmias et al., 2020). This is because in existing digital systems there must be some amount of energy paid per element-wise multiplication, which does not change with the number of multiplications, so the power must scale proportionally to the number of MACs (if other overheads are ignored) (Hamerly et al., 2019). By contrast, the multiplication in optics may be free; the energy cost is in encoding the data with enough photons such that the output signal-to-noise (SNR) is high enough for the final answer, regardless of operand size.

However, these optical neural networks (ONNs) have additional complexities and limitations of their own such as low precision, noise, and analog/digital data conversion overheads which depend on the access patterns of the model running (Figure 1). Thus, advantageously accelerating any neural network architecture with ONNs is in practice hard, and DNNs without the necessary activation statistics and model architecture may not achieve this scaling. But Transformers' efficient data-access patterns (wide layers, parallel/batched token processing, etc.), and trends in methods for scaling them, make them an especially attractive match to leverage this analog optical scaling advantage for asymptotic energy-efficiency. Here, our goal is to investigate if this Optical Transformer hypothesis is true in realistic settings — with real noise, hardware imperfections, memory and digital-analog-conversion costs, and state-of-the-art models.

Here we demonstrate how Transformers run on ONN systems, and estimate the potential benefits of doing so. To first verify that Transformers may run on these systems despite their imprecision, we sampled operations from a Transformer and ran them on a real spatial light modulator (SLM) based experimental system, and used the results to create a calibrated simulation of the optical hardware, with the systematic error, noise, and imprecision of weights/inputs we observed. Transformers running on the simulated hardware could perform nearly as well as those running digitally, and could be far more efficient. We summarize our key contributions as follows:

- We demonstrated linear Transformer operations (the bulk of a Transformer's computation) running with sufficient accuracy on real optical hardware and in a matching simulation, despite errors and noise on hardware supporting fewer than 8 effective bits of precision.

- Via simulation, we established scaling laws for optical Transformer performance versus optical energy usage, and optical energy usage versus model size. We found that Transformers accelerated optically achieve performance that is consistent with the ideal $\frac{1}{d}$-energy-per-MAC scaling possible on analog hardware, and that Transformer architectures are large enough to benefit significantly.

- Based on our simulations and experiments we estimated an orders-of-magnitude energy consumption advantage of full ONN accelerators versus state-of-the-art GPUs, exceeding $10^3$ for near-future model sizes.

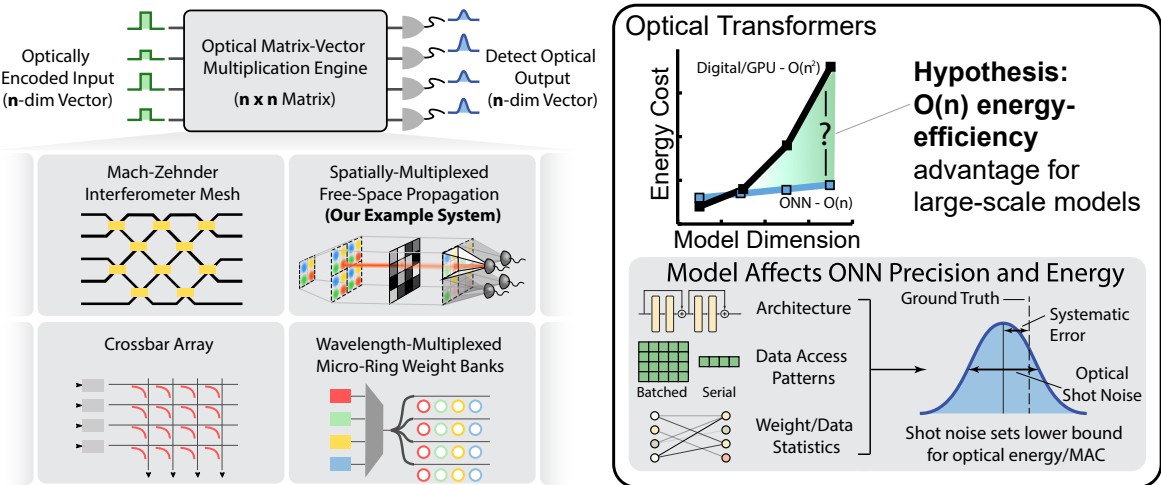

Figure 1: **Can Transformers benefit from running on optical hardware?** Left: Optical Neural Networks (ONNs) have been proposed as an alternative computing platform that can achieve asymptotic energy-efficiency advantages over digital computers running neural networks. There are various ONN platforms that all aim to efficiently implement matrix operations. Right: We hypothesize that Transformers' architecture allows for ONN-enabled benefits that scale. But energy-efficiency advantages with ONNs are not a guarantee; their behavior is affected by model architecture, statistics, and resilience to the noise/imprecision of analog hardware. Thus, while there are many implementations of general-purpose optical matrix accelerators (such as those depicted in the inset), there are still model-dependent challenges/tradeoffs in realizing their purported advantages. We seek here to answer the question of how much today's enormous Transformer models can benefit from this technology.

- We discussed how Transformers' suitability for optical acceleration is related to their architecture, and more generally how specific elements of DNN architecture affect the function of ONN systems running them.

- We identified the hardware and systems design challenges that future work on building ONN accelerators should target.

While our experiments and simulations were based on specific hardware as a representative example, our scope here is more general. We are interested in understanding how optical energy scaling and noise relate to Transformer performance and architecture. As such nearly all our findings apply broadly to linear optical processors (and hopefully future ones), irrespective of their underlying hardware implementation details (Figure 1, left).

## 2 Background and Related Work

### 2.1 Transformer Models

Transformers are models for processing sequential data based on multi-head attention. Transformers consist of two-layer feed-forward blocks and multi-head attention (Figure 2) operations. Multi-head attention computes relationships between sequence elements by deriving query, key, and value sequences $Q, K, V$ and computing dot products with a softmax nonlinearity in-between (Vaswani et al., 2017). Transformers also leverage modern design elements such as additive residual skip connections (He et al., 2016) and normalization layers (Ba et al., 2016). A defining feature of Transformers is that entire sequences may be processed in matrix-matrix products in parallel (instead of one token/input at a time).

## 2.2 Large-Scale Deep Learning

In the past few years, it has been found in particular that Transformer architectures significantly improve when sized up to billions or even trillions of parameters (Brown et al., 2020; Kaplan et al., 2020; Clark et al., 2022; Hoffmann et al., 2022; Treviso et al., 2022; Zhai et al., 2022), causing an exponential growth of deep learning compute usage (Sanh et al., 2019; Sevilla et al., 2022). These large-scale Transformers achieve ever more impressive results in not only natural language processing, but also in other domains such as computer vision (Dosovitskiy et al., 2021; Liu et al., 2021b), graphs (Kim et al., 2022), and in multi-modal settings (Jaegle et al., 2021b;a; Radford et al., 2021; Ramesh et al., 2021; Yu et al., 2022; Reed et al., 2022), making them a popular but expensive solution for many tasks—digital hardware's energy efficiency (ie. per-flop or per-inference cost) has not kept up with the growing FLOP requirements of state-of-the-art deep learning models (Sevilla et al., 2022). They also have transfer learning capabilities (Radford & Narasimhan, 2018; Devlin et al., 2019; Radford et al., 2019; Brown et al., 2020; Lu et al., 2021; Dosovitskiy et al., 2021), allowing them to easily generalize to specific tasks, in some cases in a zero-shot setting where no further training is necessary (Brown et al., 2020; Ramesh et al., 2021; Lewkowycz et al., 2022).

## 2.3 Scalable Compression and Quantization of Large Language Models (LLMs)

Optical hardware's low precision raises the question of whether scaled-up models could be quantized sufficiently to run. Thankfully, continual research in LLM compression has progressively shown that larger models do not have increasing precision requirements. For example, Li et al. (2020) found that larger Transformers can be compressed more easily, to the degree that it is more worthwhile to train large ones and compress them over training smaller ones of the target size. Furthermore, Bondarenko et al. (2021) and Dettmers et al. (2022) demonstrated running Transformers at scale with int8 precision, and the recent work of Dettmers & Zettlemoyer (2022) proposes that 4-bit is optimal for nearly all model scales, except for the largest tested (175B parameters) where 3-bit was sometimes found to work better. Some approaches utilize quantization-aware training (Jacob et al., 2018)(QAT), where a model is fine-tuned while subject to quantization, to make it robust at low precision.

## 2.4 Traits of Optical Accelerators

The typical working mechanism of optical accelerators assumed here is as follows: data, such as matrices and vectors, are encoded in light, utilizing some degree of freedom (for example, each pixel in 2d space could represent one element of a vector). This light is then modulated (such as attenuating the light) to implement element-wise products. Then the outputs are focused onto detectors, summing up the element-wise products. In essence, these accelerator systems are like a digital processor's cores (and are especially analogous to matrix compute units found on modern GPUs and accelerators) and process the data in a vectorized fashion, where at each step a batch of products (and the accumulation) happens in parallel. The difference is that the sizes that some ONN systems can process at a time can be significantly larger, such as computing products with vectors of dimension $\geq 10^3$ at a time in some cases.

Researchers have explored a wide variety of controllable optical systems to implement linear operations on optical fields, such as arbitrary matrix-vector multiplications, vector-vector dot products (Shen et al., 2017; Andregg et al., 2019; Hamerly et al., 2019; Spall et al., 2020; Bogaerts et al., 2020; Wang et al., 2022; Hayasaki et al., 1992; Mesaritakis et al., 2013; Tait et al., 2015), or convolutions (Wu et al., 2020; Feldmann et al., 2021b; Miscuglio et al., 2020; Xu et al., 2021; Fan et al., 2022). In this work, we adopt one kind of free-space multiplier (Wang et al., 2022; Spall et al., 2020; Hayasaki et al., 1992) (Figure 2, top left) to demonstrate Transformer operations in optical experiments and for our simulations. We selected this system because it has many of the same characteristics as other ONN implementations (photon detection noise, free optical data transport and reuse, systematic errors), and aim to draw conclusions that could generally be useful for those working with other ONN designs. Many ONN systems, including ours, share the following typical traits:

**Optical Shot Noise** Optical systems are subject to errors in both the actual hardware and from photon detection. Detection of optical intensity in particular is subject to a phenomenon known as *shot noise* where

the detected value is Poisson distributed: given vectors $x$ and $w$, with the elements of $x$ encoded as optical intensity, the output $Y$ is distributed as:

$$Y \sim \text{Poisson}(w \cdot x) \tag{1}$$

For other encoding schemes such as amplitude or phase encoding, equation 1 should be modified, but the detection is still subject to shot noise.

**Device Imprecision and Systematic Errors**  Systematic errors, on the other hand, are not noise but rather errors resulting from deficiencies of the hardware. Unlike noise, systematic errors are identical across multiple attempts to run the same computation. Meanwhile, because data often requires rescaling for input into analog-optical systems, neural networks running optically may encounter scaled errors. Many works studying ONNs have characterized the distribution of errors (prevalence of deviations from ground-truth values) as Gaussian (Sludds et al., 2022; Feldmann et al., 2021a).

**Free Data Transport and Reuse**  Transport and copying of data encoded in light is free when performed optically. This negates any cost of having to send data to particular sites to perform computations. Copying may be implemented in a variety of ways, such as via "fanning out" and "fanning in" data (projecting multiple copies, and then focusing multiple computation results onto a detector). However, when splitting a signal in this way, the total amount of light is divided by the number of copies.

**Efficient Photon Usage**  Shot noise, and therefore an optical dot product's signal-to-noise ratio (SNR, which serves as an effective bit precision) is related to the mean number of photons at the *output*. The efficiency of photon usage can therefore grow with increasing multiply-accumulate operations (MACs): the SNR for the product $w \cdot x$ is

$$\text{SNR}(Y) = \frac{\text{E}[Y]}{\sqrt{\text{Var}[Y]}} = \sqrt{w \cdot x} = \sqrt{\text{E}[Y]}, \tag{2}$$

which explains this behavior; if the desired output precision does not change, constant photons are required regardless of dot product size. In other words, the amount of optical energy needed is proportional to the number of vector-vector products (due to needing a certain amount of light for each), but not the amount of compute performed. For example, assume the computation of a dimension $d$ vector dot-product between two vectors. If the desired effective precision is roughly 8-bit, then one wishes to detect a maximum of roughly $255^2 = 65025$ photons at the output. If one still requires a $\sim$ 8-bit output with a dot product of size $2d$, only this same number of photons is necessary if the $2d$-sized vectors have similar statistics to the $d$-sized vectors; each element could be encoded using half the number of photons as before. Work on ONNs has studied this behavior in a variety of scenarios (Hamerly et al., 2019; Nahmias et al., 2020; Wang et al., 2022; Sludds et al., 2022).

This efficient scaling is not a guarantee—the required number of photons may be influenced by a model architecture's activation/weight distributions, encoding schemes, precision requirements, etc (Tait, 2021). Related to the previous example, if the operands of the $2d$-size dot-product have different statistics (ie. the vectors have larger dynamic ranges), or if more precision in the answer is desired for larger dot products, then differing amounts of photons are required for encoding the inputs.

## 2.5   Existing Optical Neural Network Architectures

The key similarity among systems with these traits is that they can reuse data: they accept a vector as input, but only convert it from digital signal to optical signal once to compute full matrix-vector products in the optical domain (as opposed to reloading the same data from digital-electronic memory repeatedly every time it is needed in the matrix-vector multiplication). While this reuse is achieved in different ways the concept is the same: ONN accelerators can take advantage of free data transport with optics, shot-noise-limited optical

energy usage, and methods for reusing optical data to realize an energy-efficiency advantage. Some examples of systems that possess the traits we discussed above include (see also Figure 1):

- Modulator arrays (Mesaritakis et al., 2013; Tait et al., 2016; Feldmann et al., 2021b; Giamougiannis et al., 2023): Input data is fed into a grid-like structure and routed to rows of resonators, phase-change materials, or similar elements that modulate the light, realizing a matrix-vector multiplication. Data is typically reused via the branching of the waveguides to the rows of modulating elements.

- Mach-Zehnder Interferometer (MZI) meshes (Shen et al., 2017; Gu et al., 2021): Input data is fed into a cascaded arrangement of MZIs that redistribute optical energy and information, computing a matrix-vector multiplication. The depth of the circuit allows for data flowing through to be reused (Miller, 2023) for all stages of the computation of the matrix-vector product at all depths.

- Spatial-Light-Modulator-based (SLM) ONNs (Spall et al., 2020; Wang et al., 2022): Data is fanned out, fed through a spatial light modulator that realizes element-wise scalar multiplications, and is then fanned in to compute matrix-vector multiplications. Weights may be kept in place to be used with many input vectors, which are copied via fan out.

- Fourier-domain convolution engines (Chang et al., 2018): Input data is fed through passive optical components resulting in the spatial Fourier transform; operations applied in the Fourier domain (such as multiplication by weights) thus correspond to performing a convolution. The application of weights in the Fourier domain is equivalent to reusing a spatial-domain kernel at every pixel of an image.

- Diffractive networks (Lin et al., 2018; Zhou et al., 2021; Meng et al., 2023; Zheng et al., 2023): Inputs pass through a series of diffractive elements, realizing a matrix-vector multiplication. The diffractive elements can distribute the input data in a fashion similar to MZI meshes that leverages optical depth (Miller, 2023), and weights can often be kept in place or are fixed at fabrication time.

- Frequency-domain convolution engines (Fan et al., 2022): Data is fed through electro-optic modulators (EOMs) that encode/modulate data as frequency modes, utilizing the Toeplitz-structure coupling behavior of EOMs to implement full convolutions as coupling is applied to all sets of neighboring modes automatically. This only requires loading a kernel once for it to be applied many times. Input modes can be reused by applying weights corresponding to multiple channels/features.

- Wavelength-multiplexed vector dot-product engines (Xu et al., 2021; Sludds et al., 2022): Inputs are encoded and modulated as a pulse train through EOMs, realizing vector-vector dot products by collecting element-wise multiplications at detectors. Systems can employ wavelength multiplexing so that multiple data is processed at the same time, allowing for matrix-vector multiplication. Copying data with added delay allows for convolutions (Xu et al., 2021).

While the individual implementation details are complex, what is most important are the shared traits: free data transport, analog noise/error (including optical shot noise), and linear operation computation (ie. ability to perform matrix products). Using these high-level assumptions we aim to model the behavior and efficiency of Transformers running on ONNs in general.

Previous work has considered deep learning models such as MLPs and convolutional networks on benchmark tasks like MNIST (Miscuglio et al., 2020; Wang et al., 2022), and simulations of convolutional models such as AlexNet (Krizhevsky et al., 2012) on more difficult datasets such as ImageNet (Hamerly et al., 2019). This is important in understanding the viability of these systems for low-power and edge applications, but also begs the question of how well newer, larger models perform on optical systems. Here we study Transformers running on ONN hardware to understand the operation of ONNs at compute scales that are orders of magnitude larger than previously considered.

### 2.6 Optical Neural Network Energy Calculation

**Streaming Weights or Weights-In-Place** There are two approaches for loading weights. *Weights-in-place* schemes involve loading them once, and re-using them for many inputs. Alternatively, systems can employ *streaming weights* where at every computation the required weight matrix is loaded. Streaming weights systems may be advantageous in situations where both operands of a matrix product are changing, such as in attention, or when weights are too large to be maintained by a weight-stationary device all at once; in such cases the weights would need to be reloaded for a weights-in-place system which are typically not optimized for doing so.

**Estimating ONN System Energy Consumption** ONNs' energy consumption is modelled as follows: the energy cost is broken down into the optical costs of performing MACs and the electrical costs of loading/detecting data, which are usually dominant. Consider a product between two matrices, $A \in \mathbb{R}^{n \times d}$, $B \in \mathbb{R}^{d \times k}$. Such a product results in loading (detecting) $nd + dk$ ($nk$) scalars, and performing $ndk$ MACs. If the energy to electrically load (detect) a scalar is $E_{\text{load}}$ ($E_{\text{det}}$), and to perform a MAC optically is $E_{\text{optical}}$, then the total energy is:

$$E = (nd + dk)E_{\text{load}} + nkE_{\text{det}} + ndkE_{\text{optical}} \tag{3}$$

For weights-in-place systems, one of the operands' loading costs can be assumed to be free, but in some systems maintenance of the weights which can be modelled as a small cost per MAC (given a certain throughput rate), $E_{\text{maintain}}$. Data access costs typically remain dominant due to the high costs of DAC/ADC. The calculation is then as follows:

$$E = ndE_{\text{load}} + ndkE_{\text{maintain}} + nkE_{\text{det}} + ndkE_{\text{optical}}. \tag{4}$$

This illustrates how ONNs may have asymptotic energy advantages over digital computers. Notice that regardless of the number of reuses, all data is only loaded once in Equation 3 (and partial products are accumulated at a detector before converting and storing the data digitally). Meanwhile, $E_{\text{optical}}$ ideally scales as $1/d$. These properties make energy cost disproportional to the number of MACs, $ndk$ (assuming negligible $E_{\text{maintain}}$ for Equation (4), which it typically is, and in some architectures it is zero). In other words, $\frac{E_{\text{digital}}}{E_{\text{ONN}}} \sim \min(n, d)$. For weights-in-place operations, the energy advantage scales as $\frac{E_{\text{digital}}}{E_{\text{ONN}}} \sim d$ because the weights may be reused for free.

In general, many ONN accelerators share the same approach to data processing: data is read from memory, converted to optical signal, operated on by an optical system, converted back to digital, and stored. Estimates for energy costs follow the typical breakdown (Wang et al., 2022; Hamerly et al., 2019; Chen et al., 2023): The energy $E_{\text{load}}$ is broken down into three components, related to the energy of the cost of reading from memory $E_{\text{read}}$, digital-to-analog conversion (DAC) $E_{\text{DAC}}$, and modulation to generate the light $E_{\text{mod}}$:

$$E_{\text{load}} = E_{\text{read}} + E_{\text{DAC}} + E_{\text{mod}}. \tag{5}$$

Detection energy consumption $E_{\text{det}}$ can broken down in a similar fashion, where

$$E_{\text{det}} = E_{\text{detector}} + E_{\text{amp}} + E_{\text{ADC}} + E_{\text{write}} \tag{6}$$

represent the costs of detecting a signal, amplifying the detected signal, performing analog-to-digital conversion, and writing to memory respectively. Often, the goal is to amortize these data-access-related costs via the data reuse when computing with large operands.

**Importance of Neural Network Architecture** In practice, achieving an efficiency advantage with ONNs is dependent on the neural network architecture being run. Data access (storage) from digital memory occurs before (after) digital-analog (analog-digital) conversion, so the costs for loading/storing from digital hardware are only paid once, since the optically-encoded data is freely transported/copied. Thus, the more compute to be performed per data access (ie. how large matrix-vector products are, which depends on the

DNN architecture), the more efficient the ONN. Also, the precision requirement must not change when the model being run is scaled. If this is true, then under shot noise, only the same mean number of photons at the output of a dot product is necessary, no matter how large the computation is.

## 2.7 Optical Accelerators for Deep Learning: Key Differences From Digital Compute

We use the common behaviors in ONN implementations to define a high-level abstraction of generic ONN behavior, which can be targeted by deep-learning applications. Some aspects are like those of digital accelerators. For example, ONNs access digital-electronic memory and ONN devices may only support low bit precision. But they also have unique behaviors of their own which must be considered:

- **Additional sources of error**. While ONN accelerators may operate at low precision like their digital counterparts, they also suffer from random noise and systematic errors (Section 2.4) which have different profiles/distributions from standard quantization errors.

- **Compute is effectively free**. Unlike digital accelerators which consume power to compute floating-point operations (such as multiplying two numbers), such operations can happen passively in analog, optical systems. The main cost per operation is then the cost of encoding the data, which may scale asymptotically better than digital compute in the ideal case (Section 2.4).

- **High data-access overhead**. While data access costs exist in digital-electronic accelerators, the need for analog-digital (and vice-versa) conversion makes the overhead larger in ONNs (Section 2.6).

- **Free data reuse**. While digital computers cannot freely copy data, an ONN may retrieve a vector from memory once, and use it for many dot products in, for example, a matrix-vector product, at no additional cost (besides the optical encoding of the data) (Section 2.4).

- **Model statistics affect energy consumption**. In contrast to digital accelerators which typically perform computations at a certain cost for a fixed bit precision, the encoding of data in analog-optical systems is proportional to its scalar value (Section 2.4)

In practice, these high-level concepts summarize what must be accounted for in using ONN accelerators effectively, abstracting away the need to consider the low-level hardware implementation. With these assumptions, many ONNs may be thought of as generic matrix-vector-multiplication engines, as depicted in Figure 1 (left).

# 3 Optical Transformers

We sought to evaluate Transformers running in the ONN-accelerator setting of Section 2.7. To do this, we selected an example ONN accelerator system to demonstrate the common general concepts of ONNs. We designed models that are intentionally similar to other Transformers, with the goal of simulating their behavior (informed by some experimental measurements) and energy consumption on it. A summary of our approach and model is in Figure 2.

## 3.1 Architecture and Task

We created optical Transformer models with a GPT2-like (Radford et al., 2019) architecture that replaces the GELU (Hendrycks & Gimpel, 2016) activation with ReLU6, which is known to improve low-precision model performance (Krizhevsky, 2010; Howard et al., 2017; Kim et al., 2021). For language modelling, we used the raw Wikitext-103 dataset (Merity et al., 2017). The models we simulated have 12 layers (consisting of multi-head attention and feed-forward blocks), operate on a context length of 1024 tokens, use 12 attention heads, and have embedding dimension $d$ varying from 192 to 1536. The full details of the training technique, architecture, and hyperparameters are in Appendix A.

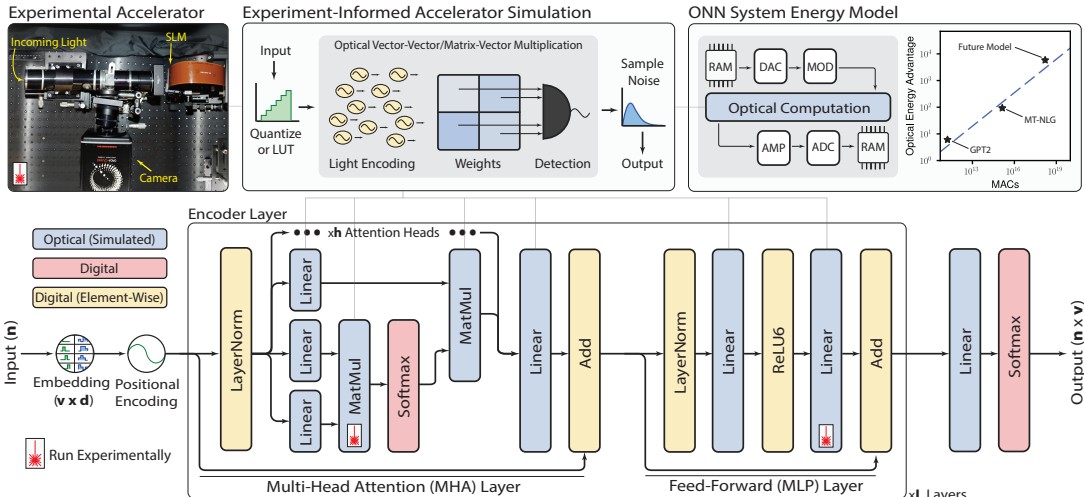

Figure 2: **Optical Transformer evaluation: prototype hardware; simulator model; Transformer architecture.** Bottom: typical Transformer architecture, but with ReLU6 activation. Top Left: experimental spatial light modulator (SLM)-based accelerator setup. From some layers—marked with a laser icon—we sampled dot products to run on real hardware. Top Middle: Linear operations, in light blue, run on a simulated accelerator with noise/error. Lookup tables (LUT) allow simulation using our setup's supported weight/activation values. Top right: our model of energy consumption for optical accelerators, based on assumptions and results from our experiment/simulations. The model accelerator system consists of random-access memory (RAM), a analog/digital conversion (DAC/ADC), light modulation (MOD), amplification (AMP).

## 3.2 Transformer Computations on Optical Hardware

We ran experiments using a real Transformer's (we used the base-sized model with $d = 768$) weights in order to characterize the behavior of an ONN system. We adopted as a representative example of an optical accelerator a spatial light modulator (SLM) based system which computes vector-vector dot products (Wang et al., 2022). Vectors are encoded on a display, and copies are shone through the SLM which has varying transmission corresponding to some data (ie. a weight matrix). The outputs of this operation—element-wise products—are collected at detectors as the resultant dot products (Figure 2, top left). We then collected calibration curves, mappings from the detected output light intensity to the actual neuron floating-point values. To do this, we ran many random dot products on the hardware and collected pairs of detected values and digitally-computed ground-truth values. We then fit the relationship linearly. We used high photon counts and averages over repeated experiments to eliminate shot noise, leaving any deviation from the linear fit as the hardware's *systematic error*.

Full details of experimental procedures and calibration are in Appendix B. There are differences between the precision limitations of real devices and linearly-spaced quantization schemes often used for DNNs - While these devices are commonly controlled by digital signals with evenly spaced discrete levels, the resultant output of these devices tends to be unevenly spaced because of their intrinsic nonlinear response or finite extinction ratios. We used lookup tables (LUT)s to model this kind of hardware error that is common to many optoelectronic devices. The LUTs were collected for the organic LED display and spatial light modulators (SLMs) by measuring levels of one device with the other at full transmission/emission. We incorporate these LUTs into both training and simulation. Backpropagation is carried out using the straight-through estimator just as for QAT, but unlike QAT once the rounding operation produces the quantized uint8 representations, the numbers are directly used to index the LUTs to produce the activations instead of dequantizing.

## 3.3 Simulation of Optical Hardware

Informed by our experiments, we constructed a simulation of the optical hardware. By simulating the hardware behavior directly we model how any arbitrary operation would behave if run on the physical setup

Table 1: Summary of simulation configurations for different evaluation and training scenarios. For simulating optical hardware we included all behaviors. For determining optical resource scaling, we focused on shot noise, and ran a plain 8-bit model for comparison.

| Setting | Op. | Shot Noise | Sys. Err. | LUT | 4-Pass |
|---|---|---|---|---|---|
| Hardware | QAT | ✗ | ✗ | ✓ | ✗ |
| Simulation | Eval | ✓ | ✓ | ✓ | ✓ |
| Optical | QAT | ✗ | ✗ | ✗ | ✗ |
| Scaling | Eval | ✓ | ✗ | ✗ | ✓ |
| Simulation | Int8 | ✗ | ✗ | ✗ | ✗ |

when it is infeasible to run large models experimentally. We aimed to emulate the noise, error, and precision that we observed in order to understand how well full Transformers would perform when running on optical hardware. The configurations for different scenarios are summarized in Table 1. We also evaluated the digital, 8-bit-QAT-trained model for comparison purposes.

**Hybrid Scheme**   Pure optical systems cannot easily compute activation or normalization functions. Thus we assumed LayerNorm, ReLU activations, and residual skip connections are performed digitally at full precision. Thankfully, even in smaller models, linear computations are the overwhelming majority (Section 4.3).

**Non-Negative Weights and Inputs ("4-Pass" Multiplication)**   An important limitation is that our display and SLM only support non-negative values. The constraint of having all-positive data is present in many but not all optical neural network systems.We worked around this by decomposing products into sums/differences of products with non-negative operands. Consider a product between matrices $W$ and $X$. If we let $W_+$ $(X_+)$ and $W_-$ $(X_-)$ be matrices with only the positive and negative elements of $W$ $(X)$ respectively, then:

$$WX = W_+X_+ - |W_-|X_+ - W_+|X_-| + W_-X_-\tag{7}$$

**Data Scaling**   On the real system, we define a maximum activation/weight value as 1.0 and minimum as 0.0. To simulate operation, the inputs and weights of every simulated NN layer are scaled to this range, and then rescaled back afterwards.

**Device Quantization**   Real hardware may only have certain number of representable levels. To emulate this behavior, we fine-tuned pretrained models using QAT and applied the following in simulation (hyperparameters in Appendix A):

- For optics-simulated layers, we applied quantization to int8 (256 levels). Then, instead of dequantizing, we used the integer values directly as indices into the LUTs that we gathered experimentally.

- We also quantized weights, but with the SLM LUT. We clamped smaller values to 0.02 in the simulation, as our SLM does not have a high extinction ratio, and the smallest transmission is 0.02.

- Accumulation can be high precision, but we used int8 quantization for outputs, since analog-digital conversion (ADC) is expensive in practice.

- We used both deterministic and stochastic rounding when quantizing, with similar results.

**Systematic Errors**   Issues like cross-talk, misalignment, defects in ONNs give rise to systematic errors. We simulated such a constraint by adding Gaussian noise to simulated model outputs, scaled relative to the mean sizes of the outputs, as this was the noise behavior we observed experimentally (it is related to the rescaling of data between 0 and 1).

**Optical Encoding and Shot Noise** We modeled optical encoding by subjecting layer outputs to simulated shot noise (Figure 2), which differs from the systematic error model. Outputs were scaled by a number such that the average photon number per feature (photons/MAC) was some target value. Each of these features was used as the mean of a Poisson distribution, which we sampled. These outputs were then scaled back down to represent neuron values. In the simulations for optical scaling we used vanilla 8-bit QAT (no LUTs or systematic error, which can overwhelm shot noise) to cleanly demonstrate the optical scaling properties—which are model-dependent and not hardware-dependent—of Transformers.

## 4 Results

### 4.1 Transformer Error Tolerance and Hardware-Simulation Accuracy

We determined experimentally that Transformer operations are able to run on real hardware without severely degraded performance from systematic errors. The bottom four panels of Figure 3 are histograms of the experimental differences from correct values. The simulated noise distributions (dotted lines) match well with the experimental data, which confirms that they are an accurate representation of the real systematic error behavior. Figure 3 (top) is a map of the performance of the simulated model over different configurations of the mean-relative (in percent) noise at every layer of feed-forward and attention blocks. The model performs well with significant noise (experimental noise levels marked with stars), within 1 perplexity from noise-free performance unless the noise is very high. While 8-bit precision was used for QAT, the optical Transformer can perform inference at lower precision, as implied by its error tolerance. We study this further in Appendix C.

### 4.2 Optical Scaling Laws

Optical Transformers achieve language modelling performance close to their digital counterparts' when shot-noise-limited at photon budgets where optical energy is negligible. The perplexities on the Wikitext-103 validation set of various optical Transformer models simulated with different total photon usage (amount used for input data) are shown in Figure 4 (left). The curves illustrate a tradeoff: larger models need larger photon totals to function well, and there are different optimal model choices based on the photon budget. We define photons/MAC as the total photon budget (amount at input) divided by total MACs. The percentage difference from the performance at 10K photons/MAC (Figure 4, middle)—chosen to represent an ideal high-precision scenario—is roughly power-law scaled in photons/MAC for all models with truncation near 10K; better performance can be had with more photons, but with diminishing returns, and the performance matches or exceeds that of the 8-bit digital models' when the photon budget is not too low ($\sim 10^2$).

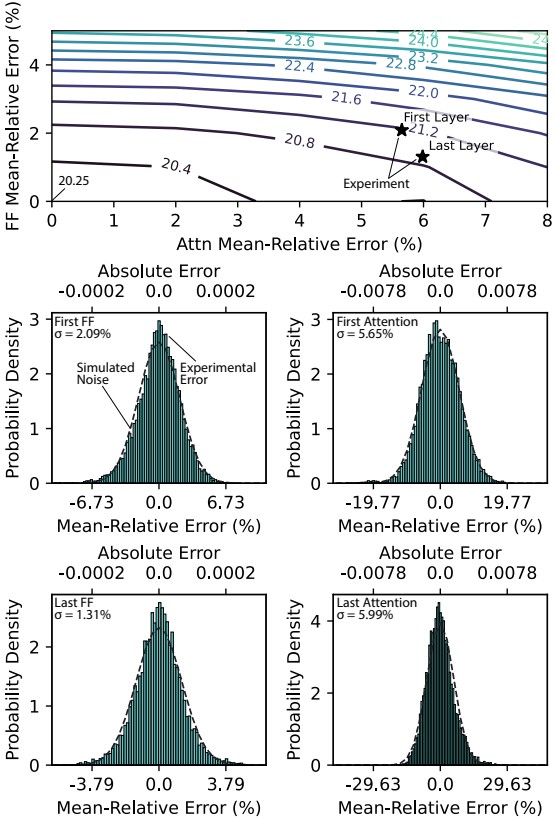

Figure 3: **Comparison of experimental and simulated noise models and simulated Optical Transformer noise tolerance.** Top: Simulated performance (Wikitext-103 validation perplexity (PPL)) versus percent mean-relative simulated noise in feed-forward (FF) and attention (Attn) layers. Systematic errors from experimental data marked with a star. Bottom: comparison of simulated noise model to error from experimental data. The Gaussian shape of the simulated error behavior matches experiment accurately.

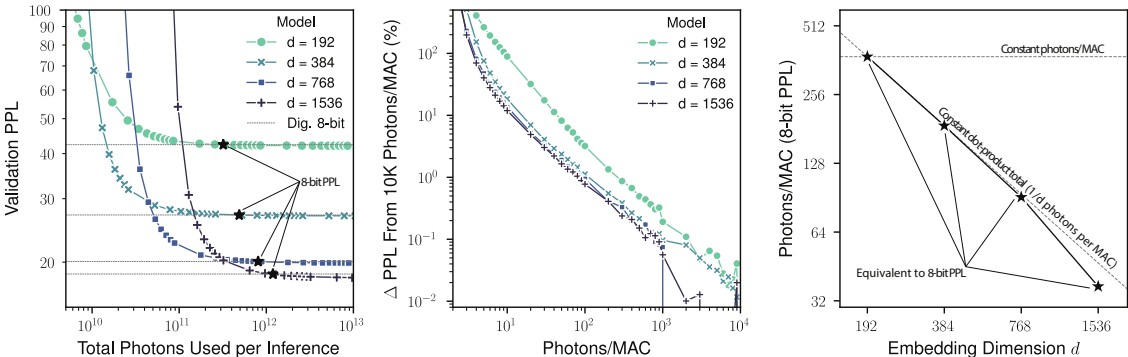

Figure 4: **Simulations of Optical Transformer behavior with varying photon usage.** Left: Wikitext-103 validation-set perplexity (PPL) versus embedding dimension $d$ and total photons used for a single forward pass/inference. 8-bit digital model performance is shown with dashed lines. Middle: perplexity degrades from ideal with fewer photons-per-MAC; the plot exhibits truncated power-law scaling. Right: Scaling of number of photons needed for an Optical Transformer to achieve the same perplexity as an 8-bit digital-electronic processor, versus model size.

The models use fewer photons/MAC as they scale, achieving the theoretical efficient scaling where the total per-dot-product photons needed is constant. To study how photon usage scales, we determined how many photons it takes to reach the performance of 8-bit digital models. These values, in Figure 4 (right), decrease nearly as $\frac{1}{d}$—the total photons needed per dot product is constant (bottom dashed line). The Transformer architecture clearly takes advantage of efficient optical scaling with larger model sizes, suggesting that required output SNR does not increase with scale. This is consistent with other work which found that Transformers compress/quantize well at scale (Li et al., 2020). Meanwhile, the already low photon usage of the largest model suggests that models larger than our simulations (>10B parameters) may use <1 photon/MAC. This sub-photon operation works in optical systems (Wang et al., 2022; Sludds et al., 2022) and is in essence no different at all from operation at higher photon counts (since the number summed at detection is still high). These empirical scaling results are tied to our specific configurations and training strategies. In Appendix H we explore a different scheme, illustrating the effects of different methods on photon usage.

## 4.3    Estimated Energy Usage

The efficient photon scaling trend we observed in Section 4.2 suggests that Transformers running on optical hardware could achieve significant energy efficiency advantages over running on digital hardware. To understand the efficiency of Transformers on optical hardware, we designed an ONN system based on current hardware that is like our experimental setup, with our measured precision and photon scaling (see: Figure 2, top right). It is an inference system with in-place weights which are loaded once and reused forever, activations read from and written to SRAM for every layer, and an optical "core" which can perform 10M multiplications per cycle (this can be thought of as a 10 megapixel SLM). We assume a 10 GHz operating speed for encoding inputs and detecting outputs, as speeds in this regime are demonstrated for high-bandwidth telecom applications and in other ONN implementations (Liu et al., 2022; Wang et al., 2019; Ashtiani et al., 2022). The photon-per-MAC scaling versus model dimension is taken to be the $1/d$ scaling which we found was possible in our simulations, and we assumed that the model operates with 5-bit input precision, 8-bit weight precision, and 7-bit output precision, as determined by our study of low precision performance in Appendix C. Energy estimates are based on the power consumption of hardware operating at this speed and precision.

Our approach to energy estimation is as follows: the system is thus assumed to have the behavior and components as described in Section 2.6; we use Equation (3) and Equation (4) to calculate the energy cost for every linear operation in Transformer models, with $E_{\text{optical}} \sim 1/d$. This includes both the model's linear layers (Equation (4), as weights are assumed to be kept in place) and the matrix-matrix products among activations in the attention operation (Equation (3), which includes the cost of loading *both* operands, as there are no static weights). The cost of digitally-run operations, such as softmax, ReLU, and other element-wise

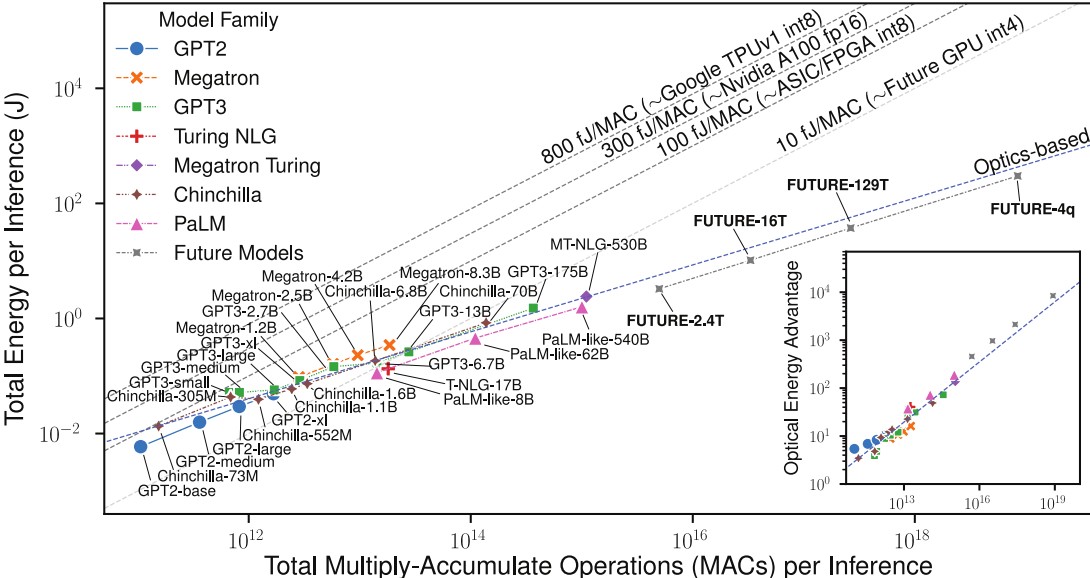

Figure 5: **Estimated energy usage of Transformer models on optical hardware for a single forward pass/inference.** Hypothetical future model designs are labelled **FUTURE-\***. Estimated energy/MAC for digital systems is based on Reuther et al. (2020). Trend for energy usage in optical systems (blue) computed based on real models only. Inset: energy advantage of running on optics over estimated NVIDIA A100 usage. The advantage grows with the model compute. $M = 10^6$, $G = 10^9$, $T = 10^{12}$, $q = 10^{15}$ parameters.

operations, is assumed to be the cost of storing and loading all operands in memory. None of the digital operations have the $\sim d^2$ scaling of the optically-run linear operations, so energy-efficiency advantages are still possible. In Appendix D we explain all assumed energy quantities, which are based on contemporary hardware.

As models grow, running Transformers on optical hardware has a large and asymptotic efficiency advantage over running on digital hardware. In Figure 5 we chart estimates of the forward pass energy required for various models[1], including a hypothetical family of large, dense Transformer models designed in a similar fashion, which we label **FUTURE-\***. For comparison, we also chart various digital systems (Reuther et al., 2020) in different performance regimes, and a hypothetical "next generation" GPU that can use $\sim$10 fJ/MAC. For small models, the optics-based system uses about the same energy, but eventually gains an advantage that scales asymptotically with the number of MACs. For the larger models, MT-NLG-530B and FUTURE-4q, the optics-based approach would have $\sim$140$\times$ and $\sim$8500$\times$ energy advantages over the current state-of-the-art GPU (NVIDIA A100) respectively.

The breakdown of compute and energy costs by source is in Appendix E. In summary we found that as models get larger the feed-forward layers require most of the computation, but that the energy of data access in attention is still very expensive due to the detection of many attention matrices across heads. Meanwhile, the costs of the digital operations become relatively small, $\leq 20\%$ of the total energy for large models, and therefore not a significant bottleneck. [2]

---

[1]The recent PaLM (Chowdhery et al., 2022) models used a modified architecture. For simpler comparison, we make our estimates using a model with GPT-like architecture but with the PaLM model dimensions, which we call PaLM-Like.

[2]Trends in the design of real models have increasingly favored optics over time. Specifically, attention loads/stores a $n \times n$ attention matrix for each of the $h$ attention heads. Models with more MLP compute per attention head have a larger overall ratio of computation to energy usage; larger $\frac{d}{h}$ is more efficient. The largest GPT2 (Radford et al., 2019) uses $\frac{d}{h} = 64$; GPT3 (Brown et al., 2020), 128; MT-NLG-530b (Smith et al., 2022), 160; and PaLM (Chowdhery et al., 2022), 384.

## 5 Discussion

The results given in Section 4.3 on optical Transformers' efficiency have implications for the design of future ONN hardware/software systems.

### 5.1 Prospects for Hardware Implementation

In Appendix G we discuss in detail the specifications for an ONN system to run large Transformers, as a target for future work in their design. In particular, we found that ONNs constructed with the following traits would be ideal:

- An efficient ONN system for Transformers must perform data copying after digital-analog conversion (fan-out) and accumulation (fan-in) of partial products before detection and analog-digital conversion.

- An ONN must perform computations with large operands in a single shot to gain an energy advantage. Once operands exceed $10^4 \times 10^4$ in size the advantage is significant, and therefore a future ONN should implement at least this level of parallelism to achieve $>100\times$ efficiency improvements over current state-of-the-art GPUs (NVIDIA A100).

- An ONN must support at least 7 effective bits of precision. The imprecision can come from various sources of noise or error as long as there are effectively $\sim 2^7$ distinguishable levels (however, recent work has demonstrated low-precision Transformers (Dettmers & Zettlemoyer, 2022; Ahmadian et al., 2023)).

- An ONN system must have sufficient fast (ie. SRAM) memory to store activations at minimum.

- An ONN should be implemented with minimal cost of its surrounding electronic components for maximum benefit. Future improvements in CMOS technology will be greatly beneficial. In Appendix F we estimate that future optics-based systems might achieve energy advantages of $>100,000\times$ running models the size of FUTURE-4q (over 300 fJ/MAC).

- An efficient ONN must encode new inputs significantly faster than weights. Since weights may be reused, encoding of the inputs may become the main bottleneck in achieving high throughput, and affects the total energy cost for weight maintenance and DAC/ADC. With our assumption of an optical processing unit computing $10^7$ products at a time, $\sim 100$ MHz speed is necessary to match the $\sim 1$ POPs performance of current GPUs. Here, we based our calculations on a $\sim 10$ GHz update rate, a speed supported by existing methods and hardware, with which significant energy-efficiency advantages can be claimed.

These generic traits/specifications are sufficient to implement Transformers efficiently, regardless of implementation details.

### 5.2 Neural-Network Architectures' Relationship to ONN Performance

The design of the software — DNN architectures, including Transformer shape and size — for these systems is also critical:

- The asymptotic advantage of optics is that once data is loaded, it may be reused N times for free, with constant energy for M-sized dot products. This suggests that architectures with large M and N benefit the most, and that wider is better than deeper when scaling a model (in terms of energy).

- The attention mechanism requires much of optical Transformers' power consumption for very little compute. Models designed with larger $\frac{d}{h}$ are therefore more energy efficient. Scaling of Transformers is conveniently following this trend already.

- The ability of Transformers to run efficiently optically is due to their parallel-processing of tokens with the same weights, and ability to tolerate the levels of noise and error present in ONN systems at scale. Thus architectures designed with similar behavior (Tolstikhin et al., 2021; Liu et al., 2021a) could also be efficient.

### 5.3 Limitations

While we have laid out the potential requirements and used simulation to predict the viability and potential gains from doing so, building a full ONN system that realizes the potential benefit is still an open challenge. For example, while optical components may perform computations cheaply and quickly, there is still the issue of supplying them with sufficient data bandwidth to fully leverage them. Integrating these components into a working system – memory, conversion, the optical elements, etc. – also presents a significant engineering challenge.

Furthermore, studying potential advantages in speed/throughput is more challenging. In this domain, different ONN implementations have different behavior, and may be optimized towards running certain neural-network architectures. Further study of the tradeoffs with speed/energy/memory could be necessary. For example, the assumed high-speed 10 GHz operation here also presents an implementation tradeoff: operating too slowly may increase weight-maintenance-per-operation costs as fewer operations are performed for constant power, but too quickly makes DAC/ADC more expensive

Finally, we note that while our estimates are for single devices, large-scale deep learning systems often consist of multiple devices working together, due to memory/compute limitations. This introduces additional data-transport costs for digital systems, and memory-bound situations affect both digital and optical systems' energy consumption in nontrivial ways. In these cases, more complicated schemes to run models efficiently are necessary, such as sharding weights across devices, and the assumption that all weights can be kept in-place must be relaxed. We further analyze this case in Appendix G. Despite these limitations, we hope that the potential benefits we studied here motivate future work in this direction.

## 6 Conclusion

We have demonstrated the ability of Transformer models to run accurately and efficiently on optical hardware through optical experiments and an experiment-informed simulation of future optical hardware. We examined Transformers' scaling behavior with optics and used our findings to show that optical systems could have a large and asymptotic energy advantage over digital-electronic ones that *grows* with the model size. For example, we showed that optical hardware may achieve an over $100\times$ energy advantage[3] when running the largest Transformer models today ($\sim$500 billion parameters) and that larger, future Transformers ($\sim$4 quadrillion parameters) may be realized with an $>8000\times$ optical energy advantage. We believe our findings about the potential energy-efficiency of optical accelerator hardware strongly motivate the development of optical processors for large-scale deep learning with Transformers.

## 7 Acknowledgements

The authors wish to thank NTT Research for their financial and technical support. Portions of this work were supported by the National Science Foundation (awards CCF-1918549 and CBET-2123862) and a David and Lucile Packard Foundation Fellowship. We acknowledge helpful discussions with and feedback from Alen Senanian, Benjamin Malia, Fan Wu, Federico Presutti, Jeremie Laydevant, Sridhar Prabhu, and Vladimir Kremenetski.

---

[3]Versus current digital-electronic hardware, which we assume has an energy efficiency of 300 fJ/MAC. This is likely a generous assumption for the digital-electronic hardware, since we do not add any energy costs for data loading and storing, whereas to compute the energy cost for optical hardware, we do count electronic memory costs—as detailed in Appendix D.

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

Table 2: Model configurations for optical Transformers. $M = 10^6$.

| Model | n | d | h | L | Non-emb. Params |
|-------|------|------|----|----|-----------------|
| Tiny  | 1024 | 192  | 12 | 12 | 15M   |
| Small | 1024 | 384  | 12 | 12 | 40.6M |
| Base  | 1024 | 768  | 12 | 12 | 123.7M |
| Large | 1024 | 1536 | 12 | 12 | 416.3M |

Table 3: Pretraining hyperparameters for optical Transformer models. All models were trained with the **AdamW** (Loshchilov & Hutter, 2019) optimizer.

| Model | Steps | Batch | lr | $\beta_1$ | $\beta_2$ | $\epsilon$ | Weight decay | Dropout | Schedule | Warmup | Stop |
|-------|-------|-------|------|-----|-------|------|--------------|---------|----------|--------|------|
| Tiny  | 90000 | 32 | 2e-4 | 0.9 | 0.999 | 1e-8 | 0.02 | 0.1 | Cosine | 2500 | - |
| Small | 90000 | 32 | 2e-4 | 0.9 | 0.999 | 1e-8 | 0.02 | 0.1 | Cosine | 2500 | - |
| Base  | 90000 | 32 | 2e-4 | 0.9 | 0.999 | 1e-8 | 0.02 | 0.1 | Cosine | 2500 | - |
| Large | 90000 | 32 | 2e-4 | 0.9 | 0.999 | 1e-8 | 0.02 | 0.1 | Cosine | 2500 | 82500 |

## A  Optical Transformer Training Hyperparameters

The optical Transformer models were pretrained on the Wikitext-103 (Merity et al., 2017) dataset and used the same tokenizer as GPT2 (Radford et al., 2019). All models used **Xavier uniform initialization** (Glorot & Bengio, 2010). The architectures are in Table 2. Embedding layers were initialized with a normal distribution with $\sigma = 0.02$. We used the AdamW (Loshchilov & Hutter, 2019) optimizer, with weight decay applied to parameters which were not embedding, gains, or biases. Dropout was applied after every linear layer (including those in attention), as well as on the attention matrix and after the softmax$(\frac{QK^T}{\sqrt{d_h}})V$ product in the attention calculation. The values of the parameters used for the training scheme are in Table 3.

After pretraining the models were quantized via our 8-bit QAT scheme. For QAT we used the RMSProp optimizer (Tieleman et al., 2012). The parameters we used for the training are in Table 4. To clamp weights and activations we employ two different approaches: first, we kept running statistics of minimum and maximum values with an exponential moving average (EMA, with parameter $\alpha$) for every layer and use those to clamp. Second, we recorded the minimum/maximum statistic throughout the network for a forward pass to apply a clipping scheme. Specifically, we clamped weights and activations to percentiles of the maximum values collected for each layer. The outputs were either rounded to the nearest integer during QAT, or stochastically rounded to nearby values. Finally, for the Base-sized model we used to run the experiments, we directly used the lookup tables (LUT) instead of "simulating" the quantization of inputs and weights (though outputs are still quantized). Table 5 details our use of these various techniques in the models.

For evaluation we used the perplexity (PPL) metric to measure the language modelling performance on Wikitext-103. We evaluated the perplexity over the entire validation set, and ran the model with context length 1024 (the same as in training) and a 1024-token stride length.

## B  ONN Experimental Procedure

### B.1  Experimental Setup

Our setup is a SLM-based matrix-vector/vector-vector multiplier. The setup is shown in Figure 6 with a simplified illustration in Figure 7, and works as follows: Vectors corresponding to the inputs and weights are rearranged into squares of pixels and loaded onto the display and SLM respectively. They are aligned such that the light from display pixels will reach the corresponding pixels on the SLM. First, light from the display enters into the polarizing beam splitter (PBS), and reaches the SLM through a half-wave plate (HWP) which rotates its polarization. The phase is then modified by the SLM and reflected back through the half-wave plate, rotating the polarization again based on the phase difference. Then, the PBS only admits light of a

Table 4: Quantization aware training hyperparameters for optical Transformer models. All models were trained with the **RMSProp** (Tieleman et al., 2012) optimizer. Quantization parameters are in Table. 5.

| Model | Steps | Batch | lr | $\alpha$ | $\epsilon$ | Weight decay | Dropout | Schedule | Warmup | Stop |
|-------|-------|-------|------|------|------|------|------|--------|------|------|
| Tiny | 7327 | 64 | 1e-5 | 0.99 | 1e-8 | 1e-5 | 0.1 | Cosine | 2500 | - |
| Small | 7327 | 64 | 1e-5 | 0.99 | 1e-8 | 1e-5 | 0.1 | Cosine | 2500 | - |
| Base | 7327 | 64 | 1e-5 | 0.99 | 1e-8 | 1e-5 | 0.1 | Cosine | 2500 | 5500 |
| Large | 7327 | 32 | 1e-5 | 0.99 | 1e-8 | 1e-5 | 0.1 | Cosine | 2500 | 5500 |

Table 5: Hyperparameters for optical Transformer Quantization. We perform QAT with both a percentile-clipping approach and by clamping based on an exponential moving average (EMA) of model statistics with factor $\gamma$. For the Base-sized model that is used in our experiments (LUT-Base), we use lookup tables (LUT) for inputs and weights instead of quantization.

| | Overall Config | | EMA | Attention Clipping | | | Feed-Forward Clipping | | |
|-------|-----------|---------------|-------|---------|---------|----------|---------|---------|----------|
| Model | Precision | Rounding | $\gamma$ | $Input_1$ | $Input_2$ | Output | Input | Weights | Output |
| Tiny | 8-bit | Stochastic | - | 99.99% | 99.9% | 99.9999% | 99.99% | 99.9% | 99.9999% |
| Small | 8-bit | Stochastic | - | 99.99% | 99.9% | 99.9999% | 99.99% | 99.9% | 99.9999% |
| Base | 8-bit | Stochastic | - | 99.99% | 99.9% | 99.9999% | 99.99% | 99.9% | 99.9999% |
| Large | 8-bit | Stochastic | - | 99.99% | 99.9% | 99.9999% | 99.99% | 99.9% | 99.9999% |
| LUT-Base | LUT | Stochastic | - | 99.99% | 98% | 99.9999% | 99.99% | 99% | 99.9999% |
| Tiny | 8-bit | Deterministic | 0.999 | - | - | - | - | - | - |
| Small | 8-bit | Deterministic | 0.999 | - | - | - | - | - | - |
| Base | 8-bit | Deterministic | 0.999 | - | - | - | - | - | - |
| Large | 8-bit | Deterministic | 0.999 | - | - | - | - | - | - |

certain polarization along one of its arms, aimed at a camera for detection. Summation of the output pixels is performed digitally. This SLM–HWP–PBS arrangement effectively creates an amplitude modulating SLM, where the output at each pixel is the element-wise product of the input pixel and corresponding weight pixel.

The OLED display has multiple color channels and a broad spectrum. For easier modulation by the SLM, we used a band-pass filter and only green light.

The components we used are:

- Organic light-emitting diode (OLED) display (Google Pixel 2016)

- Reflective liquid-crystal modulator (1920-500-1100-HDMI, Meadowlark Optics)

- Half-wave plate (PH10ME-532, Thorlabs)

- Polarizing beam splitter (CCM1-PBS251, Thorlabs)

- Zoom lens for imaging onto SLM (Resolv4K, Navitar)

- Zoom lens and objective lens for imaging onto detector (1-81102, Navita and XLFLUOR4x/340, Olympus)

- Band-pass filter (FF01-525/15-25, Semroc)

- Camera for detection (Prime 95B Scientific CMOS Camera, Teledyne Photometrics)

This setup works as a good bench for testing the precision of optical Transformers by performing optical dot products involved in attention and feed-forward layers. Even though the optical dot products were performed one at a time, it is sufficient for showing that Transformer operations can run with the accuracy of ONNs, since matrix-vector and matrix-matrix products are merely collections of many dot products run in parallel.

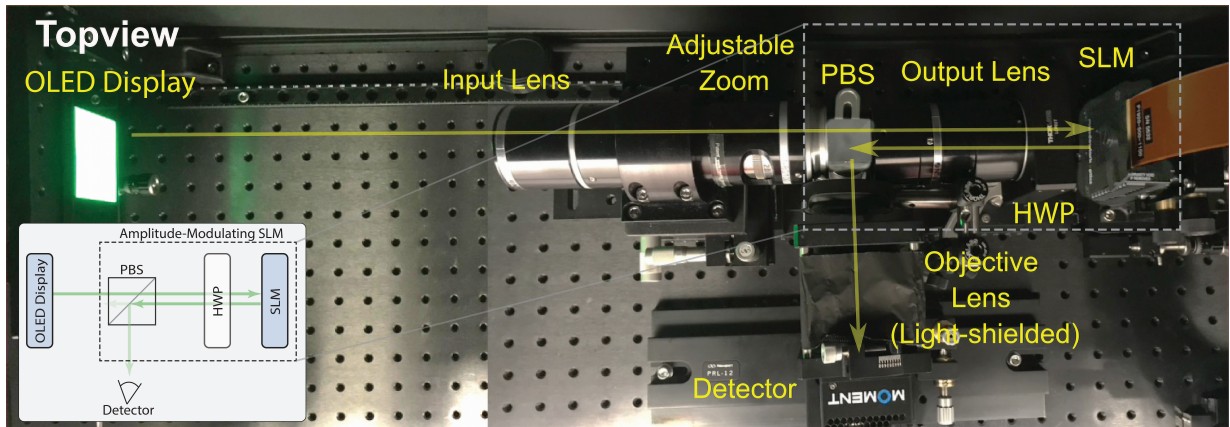

Figure 6: Photo of experimental setup used for running Transformer dot-product operations. Inset: simplified illustration of the experimental system. Spatial light modulator (SLM) + half-wave plate (HWP) + polarizing beam splitter (PBS) arrangement is effectively an amplitude-modulating SLM. The system works as follows: in our experiments, a vector is loaded as pixels on the organic light-emitting diode (OLED) display, and weights on the SLM. The input light enters through the PBS towards the SLM, passing through the HWP twice as the SLM reflects it. The SLM and HWP together rotate the polarization of the light, such that the amount reflected by the PBS towards the detector for each pixel is roughly the product between the pixel value and the corresponding weight on the SLM. The summation of these element-wise products by the detector yields the dot product.

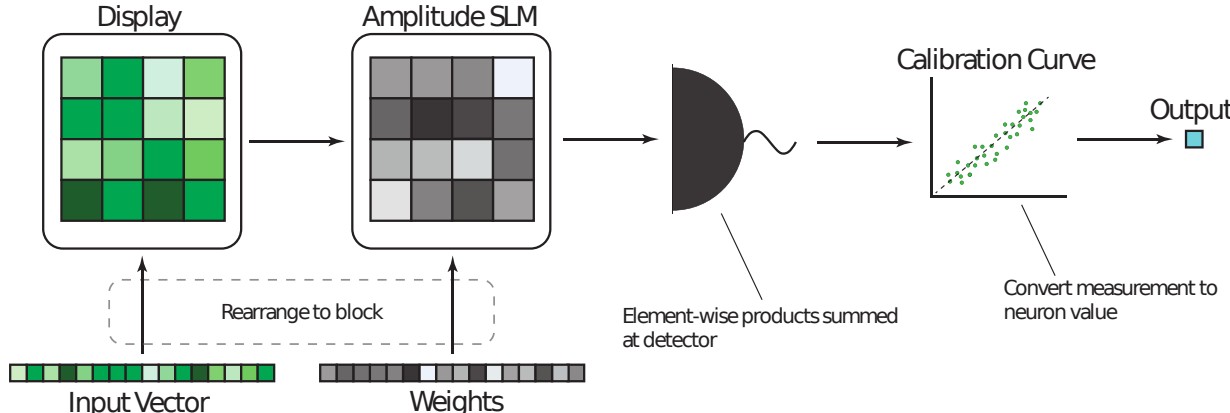

Figure 7: Simplified illustration of experimental setup operation. Weights are loaded and rearranged into a block on spatial light modulator (SLM) to prevent crosstalk between pixels of drastically different values. Data is rearranged on display accordingly. Measurements are looked up against calibration curve to obtain the final output value.

## B.2  Calibration and Lookup Tables

We used several techniques to reduce errors, map inputs to SLM/display values, and to convert detected outputs back to neural network values.

First, we developed a specialized data-pixel encoding scheme to reduce systematic errors. We noticed that a large source of error was with a limitation of our hardware—in particular the SLM pixels have cross-talk (pixels may affect their neighbors if they have very different values) and misalignment in the experimental setup may lead to corrupted outputs. To help with these issues, we created "macropixels"—each input element (and weight) does not occupy one pixel on the display (SLM) but rather is mapped to a 3x3 grid of

pixels, all with the same value. For the attention layers, we used 5x5 macropixels for the results we report, but later discovered that with 3x3 the performance is essentially the same. We also rearranged vectors into square blocks of pixels so that significantly nonzero weights are nearby each other Figure 7. For the vectors to better fit in the center of the field of view (where there is less distortion/misalignment) we computed the dot products using only the 400 largest weight elements (the corresponding input elements are loaded). While this may introduce some inaccuracy in the final results, we found that the benefits of computing the element-wise products more accurately outweigh the drawbacks of pruning the weights; the outputs were still quite accurate to the ground-truth dot-product values (see main text, Figure 3). We suspect that this was the case because:

- Transformer weights are not entirely dense; some weights were already zero.

- Because our setup only supports non-negative data anyway, we use the four-pass approach (Section 3.3, main text). This means that for any given dot product, roughly half the weights and activations will be zero before considering the previously mentioned sparsity.

- Meanwhile, a second consequence of this four-pass approach is that roughly half of activations will be zero as well, possibly rendering some of the pruned weights irrelevant.

- Transformers still perform well when pruned, and luckily larger models can be pruned more heavily (Li et al., 2020). While our pruning method is quite basic, the number of weights pruned was light (ie. $< 75\%$) compared to what is possible with more advanced methods.

This approach was not necessary for attention calculations, since the dot products were sufficiently small to fit them entirely (64 elements).

Next, we consider the lookup tables (LUTs) of the display and SLM in the setup. In order to optimize the experimental results, the model used for experiment was trained to be aware of the realistic, discrete mappable values supported by the system. The display has a LUT with 256 unique levels (1000 levels total, but many are the same as others) and the SLM has roughly 128 unique levels (256 total). So they are roughly capable of 7 and 8 bit precision. The SLM also cannot fully extinguish input light—the minimum modulation is 2% of the maximum transmission. Thus, the minimum absolute values of the weights were mapped to 0.02 instead of 0.

After applying these approaches, we finally collected the calibration curve, which maps the output intensity measurements to neuron values in the neural network and allows us to determine the experimental setup's systematic error. To do this we sampled randomly inputs and outputs of the layers we wished to run, computed their dot products both digitally and in experiment, and created a data set of experimental measurements and ground-truth-digital dot-product outputs. We then performed linear regression to find a mapping between experimental output and the correct values, effectively creating another lookup table. Then when future dot products were computed experimentally, the output was passed to this linear regression model (or it can literally be stored as a lookup table) to get the output. We used many photons and averaged outputs across multiple shots for each input, eliminating shot noise—any remaining error in this calibration scheme we defined as the system's systematic error.

It is important to note that in general other optical systems might have different causes of error from ours, but the overall accuracy of our system is representative of a typical ONN nowadays.

### B.3    Model Design Optimization

Transformers tend to have large dynamic ranges in their activations and weights (Bondarenko et al., 2021). In particular, we found that systematic error is proportional to some characteristic amplitude of the output. So, because it scales roughly with the sizes of outputs, having large outlier values can increase the systematic error and worsen the calibration for all other values in the representable range. Furthermore, after quantization in a naive, linear scheme, large outliers mean that huge ranges of outputs which are seldom used are assigned to many of the quantization levels, while the rest of the small, common outputs are squashed into few buckets—so the model precision is poor. This can be an issue when quantizing any deep learning model, but

Table 6: Simulated optical Transformer precision ablation. Input precision is degraded by subsampling from lookup table (LUT), while output is quantized. Input precision is approximate, as LUT has 1000 levels, not 1024. Bold: most compressed model found in our ablation with performance very close to the baseline.

| Input Precision (LUT) | Output Precision | Val. Loss |
|---|---|---|
| $\sim 10$ bits | 32 bits | 3.0059 |
| $\sim 9$ bits | 32 bits | 3.0057 |
| $\sim 8$ bits | 32 bits | 3.0054 |
| $\sim 7$ bits | 32 bits | 3.0039 |
| $\sim 6$ bits | 32 bits | 3.0034 |
| $\sim 5$ bits | 32 bits | 3.0017 |
| $\sim 4$ bits | 32 bits | 3.0111 |
| $\sim 3$ bits | 32 bits | 3.1223 |
| $\sim 5$ bits | 8 bits | 3.0032 |
| **$\sim 5$ bits** | **7 bits** | **3.0074** |
| $\sim 5$ bits | 6 bits | 3.0335 |
| $\sim 5$ bits | 5 bits | 3.3966 |

was exacerbated here by those systematic errors and the fact that the lowest levels of the weights are 0.02 and not 0.0. Therefore, we opted for an aggressive clipping scheme and the clamped activation ReLU6 when training the model to be run (Appendix A, **LUT-base** model); they reduce the dynamic range of inputs and weights and we found that they drastically improved the ONN's ability to run Transformer operations with smaller error. Having fewer values in the 0.02 bucket of the SLM LUT also improved QAT training stability significantly. Even though the non-zero light extinction at 0.02 is caused by the specific SLM in our setup, such issues may happen with other optical implementations made of elements with finite extinction or resolution, and here we described a method to mitigate such issues by modifying training methods.

### B.4 Transformer Dot Product Samples

While the speed and parallelism limitation of our setup made it intractable to run an entire Transformer model on it, we attempted to sample dot products to run that were representative of the range of possible activation/weight statistics in the model. That way, our results would be very representative of what running the full model would be like. In particular, we found two ways in which statistics throughout the model vary: the statistics change with depth (shallow and deep layers behave differently) and operation type (matrix-matrix multiplication in attention has different statistics from MLP layers). So, given our limited ability to run operations on the setup, we sampled roughly 10000 dot products from the first $(QK^T)$ attention operation and second MLP layer of the first and last encoder layers of the model. The inputs to the whole model were samples from the Wikitext-103 dataset. Our approach captures the range of statistics throughout a model's different components, over its depth, and when processing a real task's data. The second MLP layer has dot product size $4d$, making it the hardest to run experimentally.

In sampling the dot products, we tried to sample from both operands equally. For example, one could sample 1000 dot products by taking a single input vector and 1000 weight matrix vectors, and vice-versa, but choosing random vector pairs captures dot products involving different tokens and weights. This is important because Transformer output sizes, particularly the outlier activation values, are token-dependent (Bondarenko et al., 2021). To maintain this balance, we sample equal rows/columns for both operands. For attention layers we sample 100 from each; For linear layers, we sampled 56 rows from the input data and 200 columns from the weight matrix $W^T$, where the product being computed is $xW^T$.

## C Simulated Precision Ablation Study

To further study how the optical Transformer can perform inference at lower precisions, we conducted a simple ablation on the input and output precisions used at inference, on the 8-bit-QAT base-sized model

with LUT. We opted to leave the weights at 8-bit precision, since in-place weights are not a significant energy cost, and do not take more space/memory in these analog optical systems. In Table 6 is the performance of the model at lower precisions. With 5-bit input and 7-bit output precision, the model performs as well as the baseline. The reported precision values for the LUT are approximate, since the LUT has 1000 levels instead of $2^{10} = 1024$ levels.

When using the LUT, it is also not possible to directly change the precision of the input. Instead, we employed a subsampling scheme where the precision is degraded by rounding to every $n$'th integer level before using the LUT, where $n$ is a power of 2 and represents a reduction in the effective bit precision. The LUT of our display has 1000 levels, some levels have the same value, and we simulate the model without added noise. So we say that the original precision is initially *at most* 10 bits ($2^{10} = 1024$).

## D  ONN Energy Calculation

Table 7: Designs of models used for energy estimates. Transformers have embedding dimension $d$, process sequence length $n$, use $h$ attention heads, and have $L$ layers. M = $10^6$ parameters.

| Model | n | d | h | L | Parameters | Reference |
|---|---|---|---|---|---|---|
| GPT2 | 1024 | 768 | 12 | 12 | 117M | Radford et al. (2019) |
| GPT2 | 1024 | 1024 | 16 | 24 | 345M | |
| GPT2 | 1024 | 1280 | 20 | 36 | 762M | |
| GPT2 | 1024 | 1600 | 25 | 48 | 1.5B | |
| Megatron | 2048 | 1536 | 16 | 40 | 1.2B | Shoeybi et al. (2019) |
| Megatron | 2048 | 1920 | 20 | 54 | 2.5B | |
| Megatron | 2048 | 2304 | 24 | 64 | 4.2B | |
| Megatron | 2048 | 3072 | 32 | 72 | 8.3B | |
| GPT3 | 2048 | 768 | 12 | 32 | 125M | Brown et al. (2020) |
| GPT3 | 2048 | 1024 | 16 | 24 | 350M | |
| GPT3 | 2048 | 1536 | 16 | 24 | 760M | |
| GPT3 | 2048 | 2048 | 24 | 24 | 1.3B | |
| GPT3 | 2048 | 2560 | 32 | 32 | 2.7B | |
| GPT3 | 2048 | 4096 | 32 | 32 | 6.7B | |
| GPT3 | 2048 | 5140 | 40 | 40 | 13B | |
| GPT3 | 2048 | 12288 | 96 | 96 | 175B | |
| Turing-NLG | 1024 | 4256 | 28 | 78 | 17B | Rosset (2020) |
| MT-NLG | 2048 | 20480 | 128 | 105 | 530B | Smith et al. (2022) |
| Chinchilla | 2048 | 640 | 10 | 10 | 73M | Hoffmann et al. (2022) |
| Chinchilla | 2048 | 1024 | 16 | 20 | 305M | |
| Chinchilla | 2048 | 1280 | 10 | 24 | 552M | |
| Chinchilla | 2048 | 1792 | 14 | 26 | 1.1B | |
| Chinchilla | 2048 | 2048 | 16 | 28 | 1.6B | |
| Chinchilla | 2048 | 3584 | 28 | 40 | 6.8B | |
| Chinchilla | 2048 | 8192 | 64 | 80 | 70B | |
| PaLM-like | 2048 | 4096 | 16 | 32 | 8B | Chowdhery et al. (2022) |
| PaLM-like | 2048 | 8192 | 32 | 64 | 62B | |
| PaLM-like | 2048 | 18432 | 48 | 118 | 540B | |
| **FUTURE** | 2048 | 40960 | 80 | 120 | 2.4T | This work |
| **FUTURE** | 2048 | 81920 | 128 | 200 | 16T | |
| **FUTURE** | 2048 | 163840 | 160 | 400 | 129T | |
| **FUTURE** | 2048 | 655360 | 512 | 800 | 4q | |

The models we used to estimate the energy use of ONN systems are in Table 7. We used a variety of real models that have been introduced by other works, and then designed our family of hypothetical future models

**FUTURE-\*** in a similar fashion, keeping a reasonable sequence length, increasing the embedding dimension drastically, and following the trend of recent large models like PaLM (Chowdhery et al., 2022) and MT-NLG (Smith et al., 2022) of increasing the ratio $d/h$, which results in favorable energy calculations due to the lower fraction of memory operations in attention.

The calculation of energy costs for ONNs requires consideration of the entire system design and the costs of the surrounding electronics—since the optical computation itself is so cheap the electronics account for nearly all of the energy cost. The way the energy is accounted for is as follows: The energy $E_{\text{load}}$ can be broken down into three components, related to the energy of the cost of reading from memory $E_{\text{read}}$, digital-to-analog conversion (DAC) $E_{\text{DAC}}$, and modulation to generate the light $E_{\text{mod}}$:

$$E_{\text{load}} = E_{\text{read}} + E_{\text{DAC}} + E_{\text{mod}}. \tag{8}$$

Detection energy consumption $E_{\text{det}}$ can broken down in a similar fashion, where

$$E_{\text{det}} = E_{\text{detector}} + E_{\text{amp}} + E_{\text{ADC}} + E_{\text{write}} \tag{9}$$

represent the costs of detecting a signal, amplifying the detected signal, performing analog-to-digital conversion, and writing to memory respectively. There is also a cost of maintaining the weights in a weights-in-place system, which we call $E_{\text{maintain}}$. Because this cost scales per element, it is a per-MAC cost. But based on values from efficient commercial SLM systems, it is sufficiently small (and amortized by a large clock rate) that even the largest models we do estimations for are not bottlenecked. For optical energy, we take $1\,\text{eV}$ (single-photon energy at $1240\,\text{nm}$). We started with using our measured 8-bit-performance photon count of 1500/MAC for the smallest model ($d = 192$) and rescaled the value for larger ones using the constant-per-dot-product trend which we know our simulated models can match or beat.

The assumptions we used were that weights would be loaded from off-chip memory like DRAM (in the case of a chunked-weights strategy; for a full weights-in-place, one-shot approach this cost does not exist), and that the system uses large amounts of SRAM for activations (Fu et al., 2021). We assumed that the system only needs 5 bits worth of input precision and 7 bits worth of output precision, per the results of our ablation on the base-sized model. We still assumed 8-bit memory accesses for convenience. The actual costs for the data access and weight maintenance were assumed to be these values:

- $E_{\text{read}} = 1$ pJ/bit for off-chip memory (Sze et al., 2017), and 0.3 pJ/bit for SRAM. The SRAM estimate is based on results for DNN accelerator measurements with 9.55 pJ/32-bit access (Ponzina et al., 2021; Denkinger et al., 2020), and cutting edge/near-future assumptions for data transport from SRAM/cache (Fu et al., 2021). (Jouppi et al., 2021) estimates 14 pJ per 64-bit access, or roughly 0.22 pJ/bit, for a recent TPU architecture.

- $E_{\text{DAC}} = 10$ pJ per 5-bit sample @ $10\,\text{GHz}$—this is achievable with $100\,\text{mW}$ at 30.1dB SFDR (Caragiulo et al.).

- $E_{\text{mod}} = 1$ fJ/bit @ $110\,\text{GHz}$ with thin-film lithium-niobate modulators (Xu et al., 2022).

- $E_{\text{amp}} = 2.4$ pJ per access. A transimpedance amplifier can run at 24 mW at 70 GHz (Ahmed et al., 2014). We will just assume 10 GHz. 24mW / $10^{10}$ = 2.4 pJ per element.

- $E_{\text{detector}}$ is negligible compared to $E_{\text{amp}}$. For example, Miller (2017) calculates the cost of detection as the capacitive discharge, $\frac{1}{2}CV^2$, with capacitance $C \sim 1$ fF and voltage $V = 0.5$ V. This results in <500 aJ of energy consumption per detection. The cost is therefore negligible compared to amplification ($E_{\text{amp}}$).

- $E_{\text{ADC}} = 3.17\,\text{pJ}$ per 7-bit sample. 10 Ghz, need 7-bits of precision, so 128 conversion steps per sample – Achievable with 24.8 fJ/c-s (Liu et al., 2022) (24.8 fJ $\times$ 128 = 3.17 pJ per 7-bit sample).

- $E_{\text{write}} = E_{\text{read}}$. Actually, write access was measured to be cheaper than read access in Denkinger et al. (2020), but we use $E_{\text{write}} = E_{\text{read}}$ as a simple, conservative assumption.

- $E_{\text{maintain}} = 0.002\,\text{fJ/MAC}$. Assuming 2W for operation of a 10MP SLM, with inputs shone at 10 GHz (each pixel performs one MAC every cycle). There is not much information SLM power consumption for maintenance of a fixed pattern on the LCD panel, though more typical LCD displays which update can operate in the $\sim$1W regime. For example, Sony Corporation consumes 30 mW with 180000 pixels, which would scale to 1.67 W with 10MP (at worst, multiple SLMs/LCDs could be used in order to scale up).

## E    Breakdown-Of-Costs For Estimated ONN Energy Usage

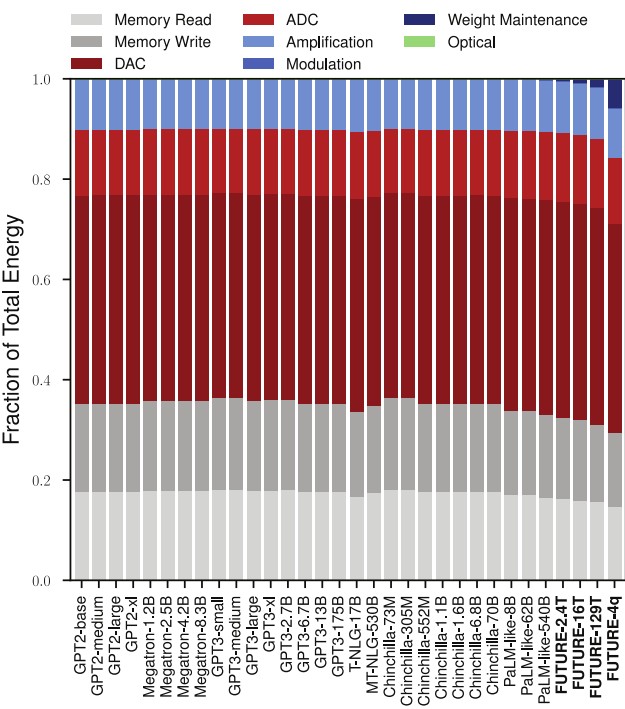

Figure 8: Breakdown of optical Transformer energy costs by energy type at 8-bit operation. Data access costs are dominant due to the high costs of DAC/ADC, but weight maintenance becomes important but not a bottleneck for large models.

In Figure 8 we see that data access costs, that is costs per element loaded/stored in memory, are most expensive. In particular, the cost of ADC and DAC are the leading contributors to the access costs, though since their cost is exponential in the bit precision, one might imagine that a future, optimized Transformer running at lower precision than our assumptions would have energy costs dominated by the actual SRAM memory costs. Also, for very large models, since the energy from weight maintenance scales with the number of MACs, it eventually will dominate if model sizes scale past that of FUTURE-4q. But future hardware would reduce $E_{\text{maintain}}$ through improved electronics or higher clock speeds allowing for lower energy per MAC. Finally, the contribution from optical energy is vanishingly small, a consequence of the efficient photon usage scaling that we found Transformers can leverage. Were it not for this, the cost of actually performing the MACs would be orders of magnitude larger than everything else, resulting in energy usage that scales the same way as digital systems'.

Breaking down the sources of compute and energy costs in Transformer models running optically further illustrates how aspects of model/system design affect energy usage. The breakdown of compute and energy costs by source is in Figure 9. We find that as models get larger the feed-forward layers require most of the computation, but that the energy of data access in attention is still very expensive. This is because of the need to save/load many attention matrices from memory, and the fact that a weights-in-place scheme cannot be used for the matrix-matrix products because the products are between activations. Of course, this also

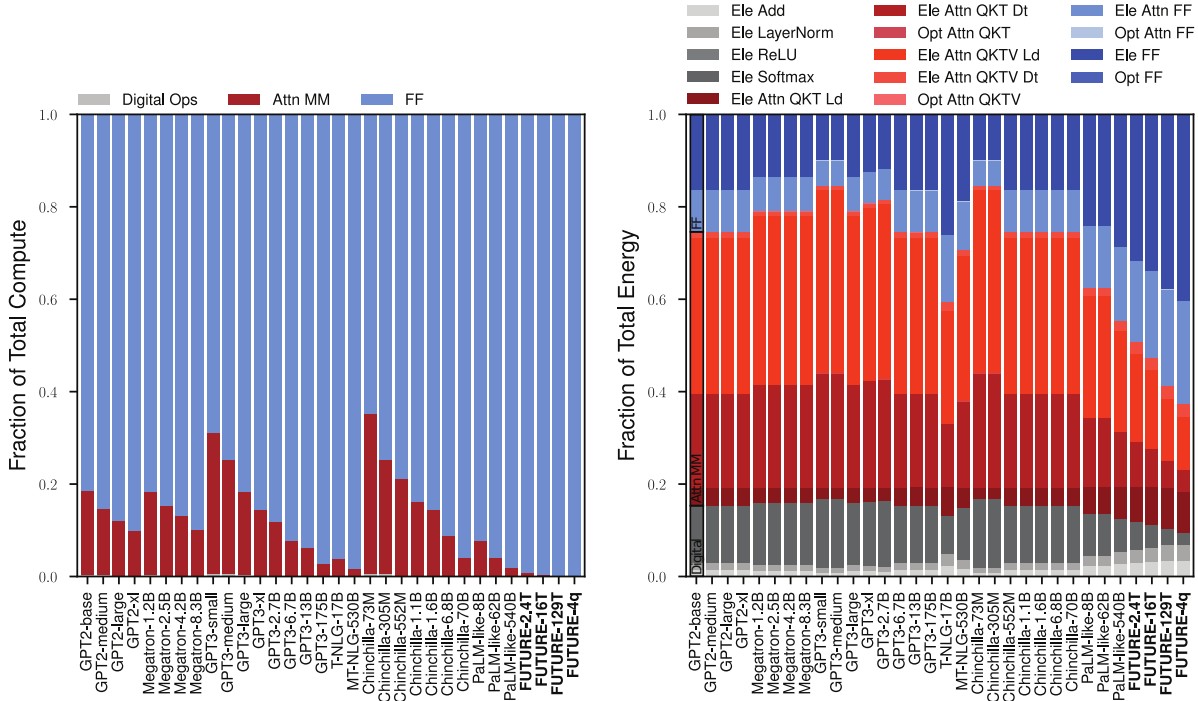

Figure 9: Breakdown of computing costs for optical Transformer models. Left: fraction of total compute used by digital operations, attention, and feed-forward components. Feed-forward layers account for most of the compute. Right: breakdown-of-costs for models by layer. The energy costs of attention operations is expensive. "Ele *" operations: electrical costs of loading (Ld), detecting (Dt), or both for data for the operation. Operations related to attention computation (ie. $QK^T$) are expensive for little compute. Functions computed digitally have their energy costs estimated as the cost of reading and writing to memory the required data.

means that there are more activations to load. In total, this means that attention layers have high energy costs for small amounts of computation. Thankfully, and interestingly, existing model design trends have moved towards focusing much harder on feed-forward layers, and so for the largest real (and our hypothetical future) models the fraction of energy cost taken by attention is low. Finally, we note that the operations we assume run on digital computers - such as nonlinear functions, in gray - do not account for much of the total energy cost (though they too are a small fraction of the total compute).

# F   Future ONN Energy Consumption

As optical accelerators are an emerging technology and as Transformer models continue to scale over time, it is worth considering how ONNs might improve over the next several years. For example, an interesting question to ask is how well future ONNs will do by the time it is possible to run a large model like FUTURE-4q. To investigate this, we estimated the energy costs of various Transformer models running optically again, but with the following changes and assumptions:

- $E_{\mathrm{maintain}} = 0$—Future weights-in-place hardware will need effectively no energy to maintain weight information (for example, one might consider the usage of phase change materials (Wuttig et al., 2017)).

- $E_{\mathrm{DAC}}$ and $E_{\mathrm{ADC}}$ are 1/32 the size—we assume that electronics could achieve a $2\times$ improvement in fJ/c-s efficiency, while future advancements in model compression allow for 4-bit Transformer models, which are much cheaper since DAC and ADC costs scale exponentially with the number of bits (Murmann, 2020).

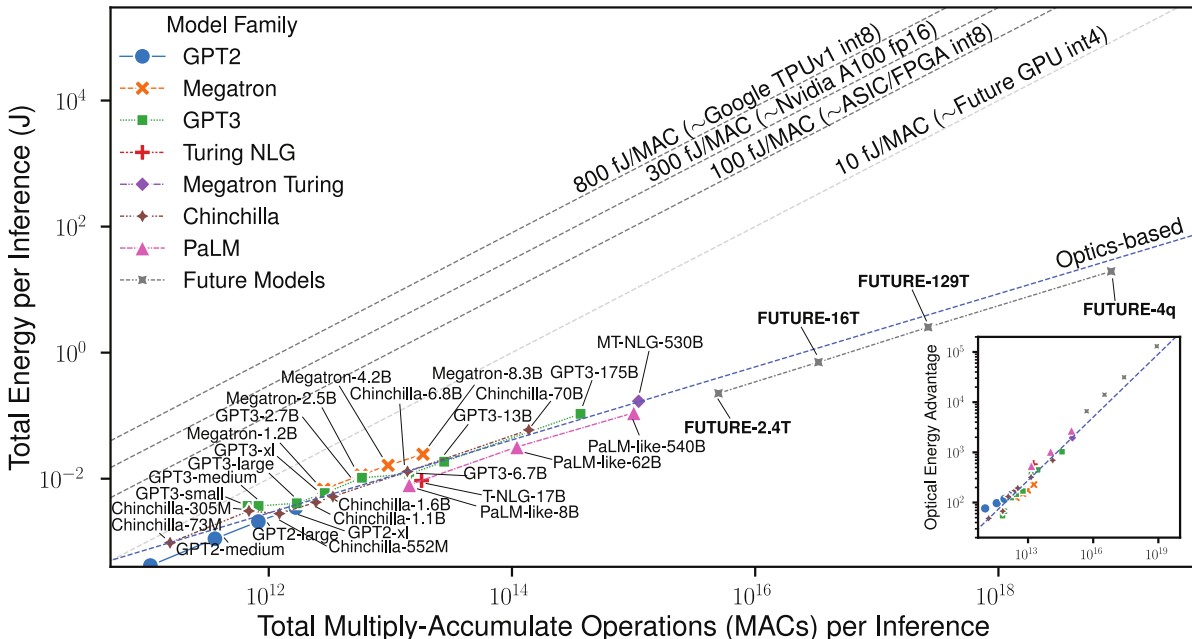

Figure 10: Energy usage estimates of forward pass for Transformers running on optical hardware, under future electronics energy cost assumptions. The energy advantages over our estimate for the current-day NVIDIA A100 GPU are larger than under our original assumptions (main text, Figure 5). $M = 10^6$, $G = 10^9$, $T = 10^{12}$, $q = 10^{15}$ parameters.

- $E_{\mathrm{read}}$ and $E_{\mathrm{store}}$ are 1/5 the size—there is already a growing recognition of the fact that AI accelerators will need high efficiency and large quantities of SRAM and DRAM in the future (Fu et al., 2021; Cerebras Systems, 2021).

- $E_{\mathrm{amp}}$ 10× cheaper (there are already cheaper trans-impedance amplifiers than our conservative estimate here, and receiver-less configuration without any amplifier has also been demonstrated (Bandyopadhyay et al., 2022)).

Under these assumptions, ONNs become far more efficient, highlighting that improvements to electronics will impact ONNs, and not just competing digital hardware. The energy scaling (Figure 10) is shifted downward for optics compared to under our previous assumptions, leading to over 1900× and 130,000× advantages over the current A100 GPU for MT-NLG and FUTURE-4q models respectively. Of course, by the time this is possible, GPU efficiency will have improved significantly as well, and we are comparing a 4-bit accelerator to the 16-bit performance of the A100. It is difficult to predict the future efficiency of GPUs at lower precision, but it is clear that ONNs can benefit from improvements to electronics and low-precision inference.

# G  Scaling ONNs: System Specifications and Communication Costs in Multi-Processor and Memory-Constrained ONN and GPU Setups

Implementation of a real ONN for large models might be difficult because the amount of hardware needed to maintain all the weights is exceedingly large. In Table 8 are the requirements for hardware to run the largest future model. To compute the number of weights/elements, we selected the largest MLP layer in the model, since that requires the most space for weights and activations. While detector and memory requirements are achievable, the number of required cores—each an optical component capable of performing 10M multiplications with weights—is enormous. There are some approaches to remedy this kind of memory issue in both GPUs and ONNs, and we are interested in their hardware-time-energy tradeoffs for ONNs.

One solution is to introduce chunking, where only a portion of the weights are loaded at a time, and the inputs are passed through. Then, the amount of time it takes to run is increased by a factor of the number of chunks. This also impacts the optical system's energy advantage over digital ones in two ways. First, the weights must be loaded, but the cost can be amortized via reuse with batched inference. This comes at the expense of latency. This is a new kind of tradeoff, since digital systems cannot reuse weight data for free. Second, for each weight chunk, all inputs must be reloaded; changing the chunk number trades energy efficiency for lower hardware requirements. These energy tradeoffs are illustrated in Figure 11; other factors dominate energy usage until the chunk number is large and chunking becomes the bottleneck.

Realizing large models with GPUs will likely also require a multi-GPU strategy, which will incur overhead over the peak performance of a single GPU. We find that with a simple model of communication costs—modelling the activation reloading in both GPU and ONN systems—that ONNs can retain some of their advantages, dependant on how much system memory (or maximum number of weight elements) is available per-processor. We created a simple model to estimate the cost of this approach in GPU systems. In GPU systems, instead of splitting a model over time, the model may instead be split over multiple GPUs. This introduces an analogous tradeoff to the activation reloading in ONNs due to communication costs: if each GPU holds some chunk of weights, then after every layer, the outputs of multiplying the inputs with each chunk must be broadcasted to every GPU in an all-to-all fashion. This is in essence an all-reduce operation—after every layer, the outputs from all GPUs must be copied onto all GPUs. In total, this means the total number of activations is loaded $k$ times, where $k$ is the number of GPUs. As a crude but conservative estimate of these costs, we modeled this by taking the cost of running the entire model on one GPU, and then adding the energy cost of loading the activations from DRAM, multiplied by the number of chunks (GPUs). This is likely an underestimate, as broadcasting data across GPUs in a real setup requires sending data electronically over much longer distances than required for DRAM access, which would be expensive.

To determine the number of chunks, we tested multiple assumptions about device memory. We assumed a value for the amount of memory that can be used to store weights and take the total number of weights for each model divided by this memory capacity to determine the number of chunks to be used.

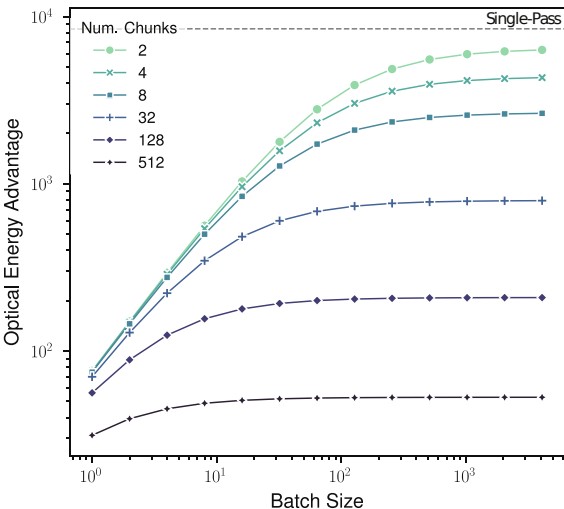

Figure 11: Optical energy advantage vs A100 (FUTURE-4q). When chunking, the cost of loading weights is amortized by increasing batch size, but the overall performance is limited by large numbers of chunks because of input data reloading.

With these models, we found that too much chunking is detrimental to ONN performance, but that there is still some energy advantage to be had if it is used sparingly (Figure 11). In Figure 12 (top) are the energy cost estimates assuming a fixed memory of 100M weights (ie. 100MPixel SLM, or RAM with 100MB capacity if each weight is one byte). We assumed that for GPU, the cost of communication is at least that of DRAM-level communication due to the physical distances between GPUs. The curves for GPUs bend upward as the communication costs begin to take over, as do the largest models running optically. The ONNs still

maintain an advantage, but the advantage stops growing with model size. Looking at the energy advantage illustrates this idea more clearly: up to a certain model size the advantage is increasing, then as the model size reaches the memory limit it begins to level off, and then the advantage begins to shrink as the cost of chunking takes over. For a small range of model sizes near this peak, the advantage is maintained, suggesting that a small amount of chunking may be useful before it quickly diminishes the energy advantage.

The optimal configuration for ONNs, obviously, is to have enough memory (cores which have weights fixed in place) so that chunking is not necessary. When plotting the advantages for larger memories (and therefore fewer chunks), the advantage gets better, and larger models become worthwhile to run. In hindsight this conclusion makes sense: the benefit of ONNs is their ability to copy data ("optical fan-out") for free for parallel computation, and so reducing this in favor of repeated memory accesses removes exactly the mechanism that gives optics-based systems their advantages. This also suggests that an "optical memory" from which fixed data can be accessed for free (or significantly less than re-access through electronics) may solve this problem, allowing for more scalable ONN design without huge amounts of hardware for weights. Currently, optics still has an advantage when using multiple cores because in principle the data could be fanned out across cores, while GPUs must pay communication costs in multi-processor setups. With a fan-out/fan-in design that can collect/spread a vector across cores, the efficiency of an entirely weights-in-place system is fully that of a single, large core.

**Comparison to Language Model Caching Techniques**    Transformers running autoregressive language modelling at inference time may utilize caching techniques (such as KV-cache in attention) to speed up and save computation for inference. However, such mechanisms also use exorbitant amounts of memory, and requires offloading to off-chip memory or farther-away memory (Pope et al., 2022), each of which is far more expensive per bit than SRAM. It is difficult to estimate the energy consumption in these scenarios, but Transformers with unrestricted attention (such as for masked language modelling (Devlin et al., 2019), vision transformers (Dosovitskiy et al., 2021), etc.) must perform the full computation in a single forward pass anyway.

Table 8: Requirements for optical accelerator running feed-forward layer (embedding dimension $d$, sequence length $n$) without chunking at 8-bit precision. The requirement of many cores to maintain weights for matrix-vector products (MVM) is high, and we assume the ONN system requires static RAM (SRAM) for saving and loading activations.

| Model | Input Vector Elements | Detectors | MVM Cores ($10^7$ weights each) | SRAM (activations) |
|---|---|---|---|---|
| FUTURE-4.1q | $2.6 \times 10^6$ | $2.6 \times 10^6$ | 170,000 | 5.37 GB |
| FUTURE-129T | $6.55 \times 10^5$ | $6.55 \times 10^5$ | 11,000 | 1.34 GB |
| FUTURE-16T | $3.28 \times 10^5$ | $3.28 \times 10^5$ | 2,700 | 671 MB |
| FUTURE-2.4T | $1.64 \times 10^5$ | $1.64 \times 10^5$ | 671 | 336 MB |
| PaLM-like-540B | $7.37 \times 10^4$ | $7.37 \times 10^4$ | 136 | 151 MB |
| MT-NLG-530B | $8.19 \times 10^4$ | $8.19 \times 10^4$ | 168 | 168 MB |
| GPT3-175B | $4.91 \times 10^4$ | $4.91 \times 10^4$ | 61 | 100 MB |
| **General** | $4d$ | $4d$ | $4d^2/10^7$ | $4nd$ |

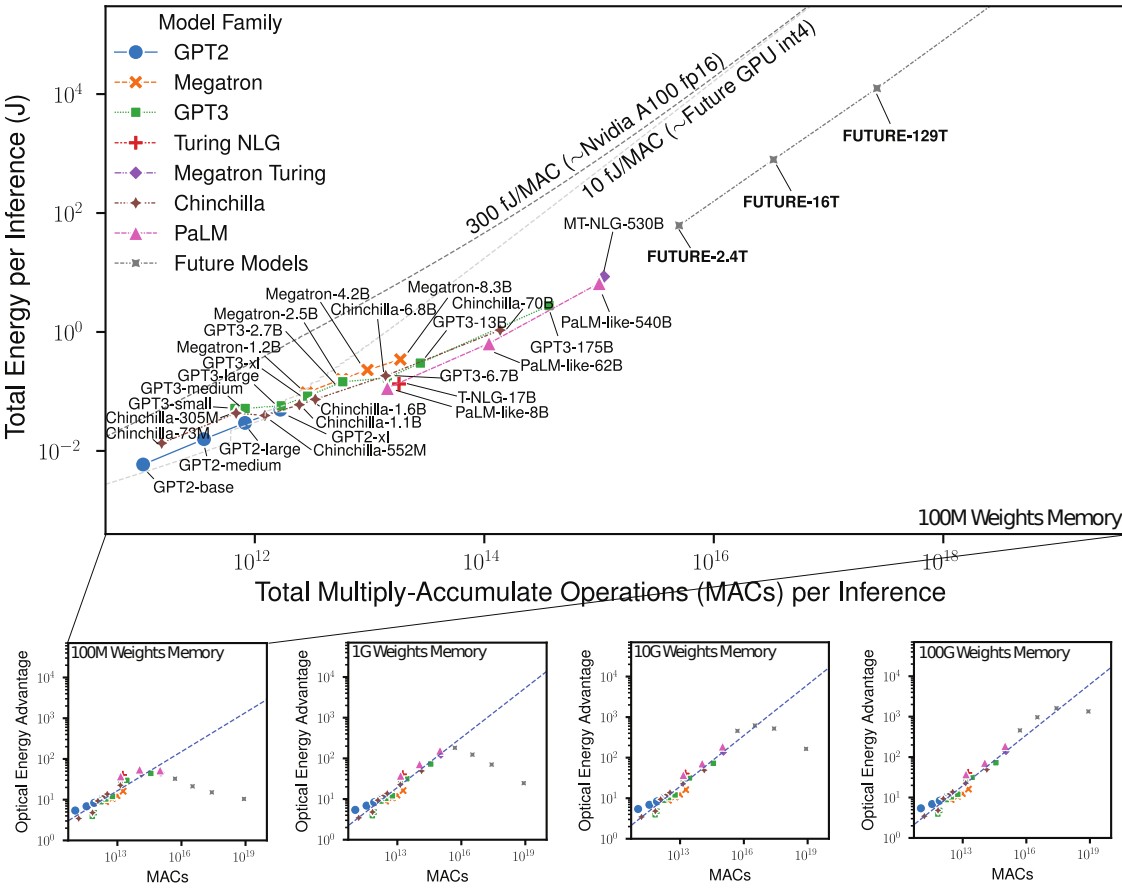

Figure 12: Energy estimates assuming a fixed processor memory size and chunking. Top: estimated energy scaling plot for Transformer models running on optical and digital hardware with 100MB of memory. As models get larger, both optical and digital systems have an upward bend in energy consumption trends, driven by communication/input-reloading-from-chunking costs. Bottom: energy advantage scaling for different memory sizes. As the memory increases, there is a maximum energy advantage for optics over NVIDIA A100 and corresponding model size before chunking costs take over. $M = 10^6$, $G = 10^9$, $T = 10^{12}$, $q = 10^{15}$ parameters.

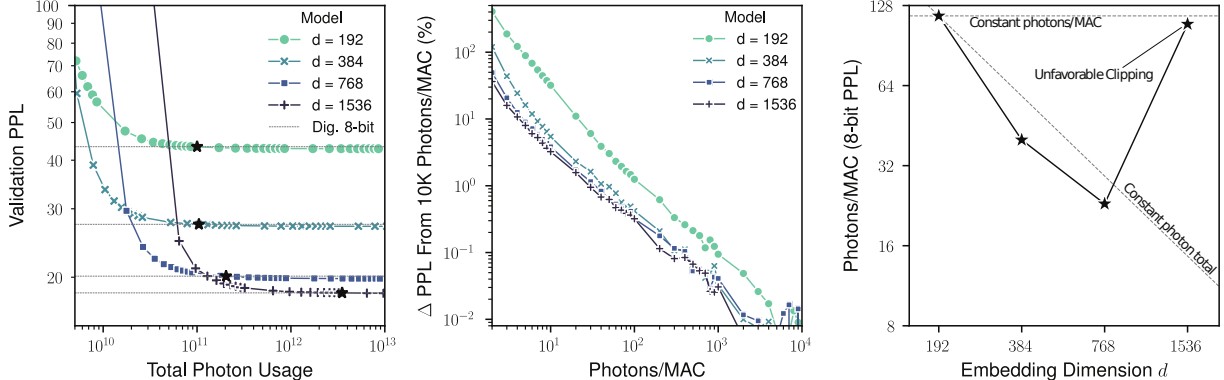

Figure 13: Behavior of optical Transformer models with varying photon usage with percentile clipping scheme. Left: Wikitext-103 validation set perplexity (PPL) versus embedding dimension $d$ and total photons usage. 8-bit quantized digital model performance levels in dashed lines. Middle: Percent change in perplexity from ideal 10000 photon count performance still exhibits truncated power-law scaling with photons per multiply-accumulate (MAC) operation for all models. Right: Scaling of photon usage for maintaining the 8-bit digital performance versus model size. Dashed lines: constant photons per dot product (optical scaling) and constant photons/MAC analogous to digital scaling. Note that unlike for our results in the main text, smaller models beat the constant-dot-product-total scaling, but the largest model exhibits poor efficiency, as the clipping scheme used here was not well suited for it.

## H   Effects of Training and Quantization Scheme on Optical Scaling

Our results demonstrating favorable scaling of photon usage in Transformers show that they can be optically efficient, but in general the photon usage is affected by the training scheme and other factors like quantization. This is because approaches for optimization quantization, regularization, etc. affect the statistics of weights and activations in the network, which unlike digital systems, are tied to the resource usage. The main example of this is with weights: they are normalized before being loaded onto an ONN accelerator, and so large outliers may lead to many weights being near 0 after normalization—admitting fewer photons through to the detector. This has a direct impact on the output SNR, and so depending on weight statistics more or fewer photons may be needed in order to run at the same precision.

To discover how a different scheme might affect photon usage, we analyzed the optical scaling of our quantized optical Transformer models with percentile clipping instead of clamping based on EMA statistics. We applied the same clipping to all models (details in Table 5). These clipped models have familiar trends in their language modelling performance versus photon numbers, but we notice key differences in the photons needed to maintain 8-bit digital performance: first, the absolute number of photons needed for the smaller models (120 and 40 versus 340 and 170 of our unclipped scheme for $d = 192, 384$) is much lower—this indicates that clipping of large weight values leads to more transmission after normalization. Second, the scaling is inconsistent, with smaller models needing significantly fewer photons than the expected $1/d$ scaling, but then requiring many photons again for the largest model. The clipping scheme degraded the performance of the large model. Of course, this could be improved by designing a better scheme for the largest model such that it requires few photons; these results illustrate how differences in the training and quantization recipe could lead to a variety of outcomes, and why efficiency is achievable but not an automatic guarantee for any scheme.

## I   Transformers With Larger Sequence Lengths

As attention is the main energy bottleneck when running on ONNs and as the demand grows for language models that can process longer sequences, it becomes important to consider the effect of sequence length in our estimates.

Table 9: Estimated energy advantage (vs Nvidia A100) of Transformers running optically with different sequence lengths ($n$).

| Model Name | $n = 2048$ | $n = 4096$ | $n = 8192$ |
|---|---|---|---|
| MT-1.2B | 8.9x | 8.9x | 8.9x |
| GPT3-6.7B | 25x | 17x | 11x |
| GPT3-175B | 73x | 45x | 27x |
| PaLM-540B | 190x | 140x | 94x |
| FUTURE-4q | 8400x | 7400x | 5900x |

Generally, we find that smaller models — which were already attention-bottlenecked — see no changes to their energy advantages, while larger ones become more attention-focused, reducing their efficiency. However, the advantages are still significant, and it depends on the architecture and number of attention heads in general. The results of estimating the energy costs of Transformers at various sizes with different sequence lengths are summarized in Table 9.

