# OpenReview forum: "Optical Transformers"
_TMLR — Accepted by TMLR_

### Review · Reviewer_VSnc · 2023-11-05

**Summary Of Contributions:**

The researchers evaluate the efficacy, throughput, and resilience of optical dot-product processors employing free-space optics to expedite Transformer computations. They present experimental data obtained from a spatial light modulator (SLM)-equipped optical setup tested across various layers within a model akin to Generative Pre-trained Transformer (GPT). The study compares the system's operational performance and computational efficiency with that of conventional digital computing systems. Furthermore, the paper discusses the expansion potential of optical processing units, underscoring the scalability prospects for optical computing infrastructures.

**Audience:**

Yes

**Broader Impact Concerns:**

no concern

**Claims And Evidence:**

Yes

**Requested Changes:**

1. How is shot noise distinguished from systematic noise in the measurements presented in Figure 3?
2. For the memory bounded settings in decoder only models such as GPT, how the proposed architecture work, given the large cost of the weight encoding cost.
3. More justification of the 10G frequency and analysis of performance sensitivity to the frequency.

**Strengths And Weaknesses:**

Strength:

The study showcases empirical findings from a free-space optical system utilizing a spatial light modulator (SLM) for performing matrix multiplication tasks. Furthermore, it explores the system's potential for scaling in conjunction with emergent technologies, highlighting the prospective advantages of optical computing in future applications.

Weakness:
1. The paper's claim to novelty is challenged due to a lack of new hardware or algorithmic advancements. The presented SLM-based optical system and its experimental use are not novel, with no bespoke hardware for Transformers and optical designs more suited for CNNs/MLPs. The algorithmic techniques mentioned are already standard in analog computing.
2. The 10G operating frequency is pretty high and could be the major bottleneck for overall energy efficiency.
3. The noise analysis concentrates on shot noise, which is minor compared to other system variations in electrical and optical components. Employing a basic Gaussian model for system errors could be an oversimplification.
4. The title is too broad given the paper only focuses on SLM-based free-space optics

---

> ### Author Response · Authors · 2023-12-11
> **Addressing weaknesses and summary of revisions**
>
> We are grateful for the reviewer's feedback and perspective. We hope to provide clarification here for the concerns raised in a point-by-point manner.
>
> Re: "paper's novelty"
>
> While it is true that there is no new algorithm or hardware, the purpose of this work was to study the behavior of Transformers in a setting where they had not been studied before. As a result we discovered:
> - We demonstrated that Transformers may run with desirable performance in the presence of optical hardware's noise and errors. This is a useful contribution in that it verifies that ONNs can be used at scale with large (Transformer) models. In general, we are not aware of previous work that has studied the behavior of ONNs when running models with the shapes and sizes of the mainstream ones deployed today.
> - We verified the "efficient photon scaling" behavior in trained Transformers. This behavior is not a guarantee for arbitrary matrix computations.
> - We predicted the potential benefits of running Transformers optically, how the benefits scale, and their relationship to the model architecture
>
> Re: "10G operating frequency"
>
> There exist a variety of existing components that can operate well within the GHz regime, used in applications such as telecom.
> - DAC/ADC can exceed 10 GHz [1]
> - Current modulators can run in the regime of 100 GHz [2]
> - 30 GHz Si-Ge photodiodes have been demonstrated in ONNs [3]
>
> Our energy calculations already include the cost of weight maintenance and the costs of energy for devices operating at this speed (Appendix E).
>
> Re: "Gaussian model for systematic errors"
>
> We believe the gaussian model to be fairly general and applicable in different ONN systems. We discussed this briefly in section 2.3: “... distribution of errors ... as Gaussian (Sludds et al., 2022; Feldmann et al., 2021a)”. Further examples include [4, 5]. Thus, it is consistent with the behavior found in other ONNs, as well as in our experiments.
>
> Re: "Title is too broad given the paper only focuses on SLM-based free-space optics"
>
> While the SLM-based system was the only one available to us for experimentation, we made sure to make claims in the article that are generalizable to other systems: “We are interested in understanding how optical energy scaling and noise relate to Transformer performance and architecture ... irrespective of their underlying hardware implementation details”. The main claims about energy advantages in particular are deliberately only based on ONNs' ability to reuse data and shot-noise-limited photon scaling, and we provided a comparison of our example system to other devices and ONN platforms with significantly different implementations, and explained how these share the same traits (Section 2.4). In terms of limitations, we claimed that an efficient ONN system for Transformers would need certain precision levels. These may not always be achievable, hence in the discussion section we laid them out based on our characterizations of Transformers on ONN hardware.
>
> Re: "How is shot noise distinguished from systematic noise"
>
> The experimental results were averaged over multiple runs, each with high photon counts. These two approaches minimize shot noise, and leave behind what we define as systematic errors. The details of the approach are described in Section 3.2. We have updated the wording to also state that we average over multiple experiments.
>
> Re: "Memory bounded settings"
>
> While detailed analysis of this case is more difficult than the main approach we took (in more memory bottlenecked settings it is hard to estimate the consumption of digital devices, since their design is proprietary and details about data-access energy costs are not publicly available to our knowledge), we study a simple version of this case in Appendix H, “Scaling ONNs: System Specifications and Communication Costs in Multi-Processor and Memory-Constrained ONN and GPU Setups”.
>
> **Revisions:**
>
> Memory bounded settings: we make more explicit reference in our main text to our analysis of this case in the appendix, and draw attention to the fact that having too little memory can be a limitation in achieving the maximum advantage with large models.
>
> Shot noise/systematic error: we updated the wording as described previously.
>
> [1] Liu et al. A 10GS/s 8b 25fJ/c-s 2850um2 Two-Step time-Domain ADC using delay-tracking Pipelined-SAR TDC with 500fs time step in 14nm CMOS technology, IEEE (ISSCC). (2022)
>
> [2] Wang et al. Achieving beyond-100-GHz large-signal modulation bandwidth in hybrid silicon photonics Mach Zehnder modulators using thin film lithium niobate. APL Photonics; 4 (9): 096101. (2019)
>
> [3] Ashtiani et al. An on-chip photonic deep neural network for image classification. Nature 606, 501–506. (2022)
>
> [4] Zhong et al. Lightning: A Reconfigurable Photonic-Electronic SmartNIC for Fast and Energy-Efficient Inference. ACM SIGCOMM. (2023)
>
> [5]  Spall et al. Hybrid training of optical neural networks. Optica 9, 803-811. (2022)

---

### Review · Reviewer_hkwB · 2023-11-15

**Summary Of Contributions:**

The paper proposes a study on optical implementations of transformers. For this purpose the authors build an optical multiplier and use it during the weight-input multiplication. The implemented model uses various optimization such as quantization, noise simulation, etc. The authors claim that the implementation when running on transformers allows to scale the energy (photonic energy) at a $\frac{1}{d}$ as compared to $d^2$ of digital systems.

**Audience:**

Yes

**Broader Impact Concerns:**

No ethical concerns in the paper are present

**Claims And Evidence:**

Yes

**Requested Changes:**

Main Changes to be done:
- The energy is explained by breaking it into two components: optical and electric. But the overall scheme is never explained. Please provide more details


English and technical details:

- Page 4, costs this often -> cost is often
- QAT is used before being defined
- Figure 1 is not well described in the caption
- Figure 2 is well described but is overused. For instance on page 8, the authors state: "We simulated such a constraint by adding Gaussian noise to simulated model outputs (Figure 2),..." I do not see how Figure 2 is helpful in this context.
- Intuition of $d^2$ scaling of digital systems should be either referenced or explained.

**Strengths And Weaknesses:**

Strengths:
- An implementation in optical hardware of a quantized ANN (Transformer)

Weakness:
- The paper seems to try to leverage transformers as a tool to demonstrate the cost effectiveness of ONNs. However what I see that this cost effectiveness is not related to transformers. Rather as authors are pointing out the cost-effeciency comes from the ration dataloading/data operations.
- The overall contribution. It is not clear to me how the contribution is significant or different enough from works already cited by the authors - Li. et al., 2020, Wang et al., 2022, etc. I think the work would be significant if the encoding would be studied at a much lower scale such as individual photons rather than a statistical average of the photons scaling.  The paper has contribution but among the provided previous research it is not clear if the observed scaling of energy  in the frameworks of transformers is a general observation. The authors claim multiple times that the efficiency depends on the model architecture however their observation is mostly done on the optical multiplication which is transformer independent.

---

> ### Author Response · Authors · 2023-12-11
> **Paper revisions and discussion of contribution**
>
> We thank the reviewer for their valuable feedback and attention to detail in providing corrections to the writing. We hope to address the weaknesses here in a point-by-point manner, and summarize our revisions to the work.
>
> Re: "cost effectiveness is not related to transformers"
>
> While it is true that the cost-effectiveness comes from data loading/operation ratios, it is the Transformer architecture that:
> 1. enables these ratios to be large enough to gain an energy advantage when running with optics, due to its approach to parallel-processing and wide layers.
> 2. has desirable performance at scale; while other models may run well on ONNs at these scales, they do not necessarily have the property where their performance on real-world tasks continues to improve significantly past the billion-parameter regime.
> 3. has the largest-scale models deployed commercially for inference
>
> Thus Transformers have emerged as the first models to have all of the most important traits that enable an advantage with optical acceleration, compared to those previously studied in the literature.
>
> Re: "overall contribution"
>
> Even though DNN models are fundamentally composed of matrix operations, the efficiency and behavior of ONNs is highly architecture-dependent [1]. For example, given a vector, if a weight matrix has a high dynamic range of values, then the number of photons admitted to the output when run optically would be very small compared to a product with a weight matrix with nearly-equal elements. Even within the case of Transformers, this means that the training/quantization scheme leading to the final model statistics can affect the scaling (we demonstrate one such case in Appendix I). Based on this intuition, we summarize our contributions:
>
> - We have demonstrated the ability of Transformers to run on optical hardware at all. Different DNN architectures have differing precision requirements and robustness to error, which is why this finding for Transformers is a new, concrete finding of our work.
> - We proved that Transformers can be run scalably using constant photons per dot product, even though this is not always possible (we mentioned the architecture-dependence of this issue in Section 2.3).
> - We built an energy estimate (by using these previous results) to project the possible energy advantage of running Transformers on ONN hardware. While this style of analysis is commonplace, we are not aware of other work presenting the results of applying this technique to Transformer models specifically or at similar scales.
>
> Re: “energy… overall scheme”
>
> We described an energy calculation scheme in subsection 2.5 as part of the background section. There, we discussed how costs are broken down into not only optical/electrical but also the various components of the electrical costs related to data access which include conversion (DAC/ADC), memory, amplification, modulation, and detection. We wish to clarify that this scheme is indeed the approach we took, as mentioned in section 4.3, but also with weight-maintenance-cost assumptions, mentioned in Appendix E. We believe that adding these weight-maintenance assumptions to the main text would complete its treatment of our approach to modeling the energy costs. We also provided a breakdown-of-costs analysis in Appendix F highlighting how parts of the scheme and Transformer architecture contribute. The specific energy quantities used are in Appendix E.
>
> **Revisions summary:**
>
> Energy calculation scheme: We have updated the text to be more clear that the approach we described is exactly the one we used later in the article. We also made sure to include discussion about the additional weight-maintenance assumption in the main text, when it was previously in the appendix .
>
> Errata/writing corrections: we appreciate the reviewer for pointing out these issues with the text and have corrected and clarified some descriptions.
>
> Explanation of digital system scaling: we added some discussion in the introduction and background sections explaining this claim. In essence, it is because in existing digital systems there must be some amount of energy paid per element-wise multiplication, which does not change with the number of multiplications, so the power must scale proportionally to the number of MACs (if other overheads are ignored). This is necessarily true because digitally multiplying two elements is a thermodynamically irreversible process [2]. By contrast, the multiplication in optics can be done for free; the energy cost is in encoding the data with enough photons such that the output SNR is high enough for the final answer, regardless of operand size.
>
> [1] Wang et al. An optical neural network using less than 1 photon per multiplication. Nature Communications, 13 (1). (2022)
>
> [2] Hamerly et al. Large-scale optical neural networks based on photoelectric multiplication. Physical Review X, 9(2):021032. (2019)

---

### Review · Reviewer_ZbAk · 2023-11-28

**Summary Of Contributions:**

The paper explores the use of optical accelerators for running Transformer models, focusing on their potential for significant energy efficiency over digital-electronic systems. It presents a study involving simulations and experiments with optical hardware to demonstrate the advantages of optical transformers. The study predicts a substantial energy-efficiency advantage for optical systems, particularly with larger Transformer models, suggesting an over 100x advantage for models with hundreds of billions of parameters and an over 8,000x advantage for future models with quadrillions of parameters. The paper concludes by underscoring the promise of optical hardware in deep learning and acknowledging the challenges in realizing fully operational optical neural networks.

**Audience:**

Yes

**Broader Impact Concerns:**

There is no broader impact concern for the paper.

**Claims And Evidence:**

Yes

**Requested Changes:**

1. For a broader machine learning audience, some parts of the paper, especially those dealing with optical hardware specifics, could be made more accessible with user-friendly explanations or analogies.

2. While the paper discusses the advantages of optical systems over digital ones, a more detailed comparative analysis would be beneficial. Currently, the paper only compares with A100. However, many existing transformer accelerators have excellent energy efficiency. It will be fair for the authors to compare their work with state-of-the-art hardware for transformers.

3. More details on the digital part of the system will help support the energy efficiency claim. It will be interesting to know the tradeoff of the system and if the digital overhead will dominate the compute cost when the dimension grows.

4. More details on how the system computes self-attention with dynamically generated matrices are needed. Particularly, the paper needs to compare the computation with the regular linear operation of a dense layer.

5. The self-attention component in the transformer will dominate the computation when the sequence of the input increases. The paper only analyzes the dynamic of energy changes regarding the network's embedding dimension. However, it will be even more interesting to see how the energy changes when the number of tokens in the sequence changes.

**Strengths And Weaknesses:**

Strength

1. This paper gives a large-scale analytical study on the potential of optical computing for Transformer models amidst the constraints of optical hardware being limited to smaller networks. This approach underscores the feasibility of optical systems for handling large-scale models and forecasts the future capabilities of optical computing. By projecting the benefits and scalability of optical hardware beyond its present limitations, the paper offers a future perspective on the evolution of computing technologies, emphasizing the anticipated shift towards more energy-efficient and powerful optical computational methods. This foresight is crucial in guiding future research and development in optical computing.

2. A key advantage highlighted in the paper is that photon usage in optical computing decreases inversely with the increase in the dimension of computation (1/d). This discovery implies that as the scale of computation grows, optical systems become increasingly photon-efficient. This efficiency is particularly significant for large-scale Transformer models, suggesting that optical computing scales well and becomes more energy-efficient with larger models. This insight into photon efficiency is a notable contribution, emphasizing the potential of optical computing for large, complex computations in a more resource-efficient manner.

Weakness

1. One weakness of the paper is its complexity and assumed knowledge level in the writing, making it challenging for readers with a limited background in optical computing. The paper delves into the technicalities of optical hardware and theoretical concepts of optical computing, which might be challenging to grasp for a machine learning audience not versed in these areas. It lacks more straightforward explanations and analogies that could bridge the gap between advanced optical computing concepts and their relevance to machine learning. This approach can make the paper less approachable and more challenging to understand for those without a strong foundation in optical computing, limiting its accessibility and potential impact on a broader machine learning audience.

2. Another weakness of the paper is its insufficient detail in explaining computations for various components within the Transformer block. Firstly, the paper falls short of clearly articulating the digital computation overhead, particularly for components that extend beyond basic matrix multiplication. This gap in information leaves readers uncertain about the system's total computational cost and efficiency. Secondly, the paper does not adequately explain the execution of multi-head self-attention, a critical aspect of Transformer models. Given that self-attention involves dynamically generated matrices, understanding its operational intricacies, especially the potential additional costs compared to computations with fixed weight matrices, is crucial. This lack of clarity on key computational processes within the Transformer framework could lead to an incomplete understanding of the model's overall efficiency and performance in an optical computing context.

---

> ### Author Response · Authors · 2023-12-11
> **Addressing weaknesses and summary of revisions**
>
> We thank the reviewer for their insightful feedback and advice regarding the writing. We hope to address the concerns raised here and in the revised article.
>
> Re: “Complexity and assumed knowledge level in the writing”
>
> We agree that some of the descriptions and characterizations of optics-related details were terse.  We have revised the article to provide more approachable examples and explanations.
>
> Re: “insufficient detail in explaining computations for various components within the Transformer block”.
>
> We apologize for the lack of clarity regarding this issue. The claim we made was that all digitally-run operations are element-wise on the activations (besides softmax in attention, which still does not scale as d^2, where d is the width of the model), which means that the amount of compute performed digitally becomes vanishingly small at larger scales. But the arguments surrounding this claim were mostly presented in the appendix. We have made an effort to move the most important details to the main text so that it is clear that the digitally-run operations are accounted for.
>
> Our approach for calculating the digital overhead is as follows: because the operations perform minimal computation, they are typically memory-bottlenecked, so we approximate their cost as the cost of all memory read/write accesses for the operands, much like how we calculate the electronics data-access costs for the optical operations, but with a key difference: since the computation is happening digitally, there is no DAC/ADC to account for. A breakdown of the sources of energy costs is provided in appendix F.
>
> Re: “does not adequately explain the execution of multi-head self-attention”
>
> Because self-attention involves linear, matrix-matrix operations, we assume it is run optically too.  We calculated the energy costs while accounting for the behavior of MHA: we include the costs of loading both operands, as if in a streaming-weights setup. This makes attention more expensive than feed-forward layers where activations are multiplied with static weights  (discussed in Appendix F). Meanwhile, we account for storage costs by estimating the cost of storing each of the h attention heads.
>
> Alternatively, for large-scale models, multi-query attention [1, 2] is a strong alternative to MHA which reuses the same K and V matrices across all heads. This case is much closer to the weights-in-place scenario and could be more efficient in the way that linear layers are.
>
> Re: Currently, the paper only compares with A100
>
> The energy estimates presented are also plotted alongside estimates for more cutting-edge digital processors such as ASIC implementations. Since for digital systems we assume the energy-per-mac improves by constant factors, the advantage of our ONN estimates should simply be rescaled. For example, we estimate the ballpark consumption of ASIC/FPGA as ~100 fJ/MAC. In general these estimates are meant to be ballpark values, as it is difficult to understand in detail the effects of IO and compute workloads on proprietary commercial processors.
>
> Re: Transformer sequence length
>
> We made and included further calculations of ONN energy efficiency when the sequence length scales. Overall, the energy efficiency advantage is degraded in larger models that are dominated by MLP compute (as they become more attention-bound), but ONNs still enjoy significant and scaling advantages, and we note that large-sequence-length transformers often resort to other methods to replace vanilla attention, avoiding its intense memory usage because it is also a problem in digital systems. Meanwhile, smaller models, which are already more attention-bottlenecked, do not lose much of their advantages.
>
> **Revisions summary:**
>
> Assumed knowledge level of writing: additional explanations, including examples, comparisons to traditional digital hardware, and a summary of the most important concepts related to our claims and experiments: the data reuse, efficient photon scaling, and device error/imprecision.
>
> Digital operations computation scheme: we have included a description of our approach in the main text so that it is more obvious that the costs of these operations are included in our calculations, and that any numbers presented account for them already.
>
> Attention operations: While we reference these ideas in the text, we have updated it to make them more apparent so that it is obvious that we carefully considered attention operations in our cost estimates, as described above.
>
> Scaling Transformer sequence length: We have updated the article with calculations of energy efficiency when the sequence length of the Transformer models change (appendix J).
>
> [1] Noam Shazeer. Fast Transformer Decoding: One Write-Head is All You Need. arXiv:1911.02150. (2019)
>
> [2] Chowdhery et al. PaLM: Scaling language modeling with Pathways.  arXiv:2204.02311. (2022)

---

### Decision · Action_Editor_uEy1 · 2024-02-03

**Recommendation:** Accept with minor revision

**Comment:**

This submission explores the use of optical accelerators for running transformer models as a means of reducing energy cost of running large scale state-of-the art language models. This alternative is an interesting alternative solution for efficient inference of large foundation models.

One of the reviewers noted that the paper is good as it goes to hardware implementation. It provides enough details for reproducibility. However it was pointed out that the communication and clarity of the achievement is lacking.  Reviewer noted that the information may not be easily accessible to typical TMLR readership.

As noted in the audience section,  the action editor had difficulty identifying expertise within the reviewer pool to properly assess the merit of the submission. There may be some argument for the submission to find a more suitable venue outside of machine learning journals.

While one reviewer recommended "leaning reject" citing clarity and readability to the TMLR audience, two reviewers were "leaning accept". One of the reviewers (ZbAk) raising readability concerns, pointed out that revisions have improved readability to non optical computing experts and authors addressed most of the concerns in the review.

Action editor sees interesting promises from the submission. However, in current form the submission needs improvement for the TMLR readership. To mitigate some of remaining concerns, therefore,  AE recommends acceptance with minor revision addressing some of reviewer concerns:

1) modifying title to better represent limited scope/approach
2) more clarity and discussion on feasibility of 10G operation
3) if possible, improved clarity for broader ML audience (to reach wider readership and entice discussions)

**Audience:**

Audience is a challenging point regarding this submission; the significance of the paper is exploring optical implementation of Transformer models which is not directly geared towards the machine learning research community. While there are interesting comparisons to be made for current accelerator based methods; claim and methodology is more on the optical engineering front rather than machine learning.

Action editor had difficulty identifying expertise within the reviewer pool to properly assess the merit of the submission. There may be some argument for the submission to find a more suitable venue outside of machine learning journals.

**Claims And Evidence:**

While one reviewer (ZbAk) see the work is complete and the study can support the claims in the paper, one reviewer found the author's response on the feasibility of 10G operation and the broader claim of the title are not well addressed.

As suggested by reviewer VSNC, the title is too broad given the paper only focuses on SLM-based free-space optics and would suggest proper changes so that title represents the content inside the paper well.